*J Physiol* 602.23 (2024) pp 6531–6552 6531

# Circadian patterns of behaviour change during pregnancy in mice

Georgia S. Clarke[1,2,3] (ID), Andrew D. Vincent[4], Sharon R. Ladyman[5,6], Kathryn L. Gatford[1,2,3] (ID) and Amanda J. Page[1,2] (ID)

[1] *School of Biomedicine, University of Adelaide, Adelaide, South Australia, Australia*
[2] *Nutrition, Diabetes & Gut Health, Lifelong Health Theme, South Australian Health and Medical Research Institute, Adelaide, South Australia, Australia*
[3] *Robinson Research Institute, University of Adelaide, Adelaide, South Australia, Australia*
[4] *Freemasons Centre for Male Health & Wellbeing, Adelaide Medical School, The University of Adelaide, Adelaide, South Australia, Australia*
[5] *Centre for Neuroendocrinology, School of Biomedical Sciences, University of Otago, Dunedin, New Zealand*
[6] *Department of Anatomy, School of Biomedical Sciences, Dunedin, New Zealand*

Handling Editors: Paul Greenhaff & Josiane Broussard

The peer review history is available in the Supporting Information section of this article (https://doi.org/10.1113/JP285553#support-information-section).

*The Journal of Physiology*

**Abstract** Food intake and activity adapt during pregnancy to meet the increased energy demands. In comparison to non-pregnant females, pregnant mice consume more food, eating larger meals during the light phase, and reduce physical activity. How pregnancy changes the circadian timing of behaviour was less clear. We therefore randomised female C57BL/6J mice to mating for study until early ($n = 10$), mid- ($n = 10$) or late pregnancy ($n = 11$) or as age-matched, non-pregnant controls ($n = 12$). Mice were housed individually in Promethion cages with a 12 h light–12 h dark cycle [lights on at 07.00 h, Zeitgeber (ZT)0] for behavioural analysis. Food intake between ZT10 and ZT11 was greater in pregnant than non-pregnant mice on days 6.5–12.5 and 12.5–17.5. In mice that exhibited a peak in the last 4 h of the light phase (ZT8–ZT12), peaks were delayed by 1.6 h in the pregnant compared with the non-pregnant group. Food intake immediately after

The Journal of Physiology

dark-phase onset (ZT13–ZT14) was greater in the pregnant than non-pregnant group during days 12.5–17.5. Water intake patterns corresponded to food intake. From days 0.5–6.5 onwards, the pregnant group moved less during the dark phase, with decreased probability of being awake, in comparison to the non-pregnant group. The onset of dark-phase activity, peaks in activity, and wakefulness were all delayed during pregnancy. In conclusion, increased food intake during pregnancy reflects increased amplitude of eating behaviour, without longer duration. Decreases in activity also contribute to positive energy balance in pregnancy, with delays to all measured behaviours evident from mid-pregnancy onwards.

(Received 24 August 2023; accepted after revision 22 February 2024; first published online 13 March 2024)

**Corresponding authors** A. J. Page & G. S. Clarke: Vagal Afferent Research Group, School of Biomedicine, University of Adelaide, Adelaide, SA 5000, Australia.     Email: amanda.page@adelaide.edu.au & georgia.clarke@adelaide.edu.au

**Abstract figure legend** We aimed to determine the impact of pregnancy on the timing of food and water intake, activity and sleep in mice. Food and water intake was increased in late pregnancy, with a 1.6 h delay in the peak in food intake late in the light phase and 0.34 h delay in peak water intake late in the dark phase in comparison to the non-pregnant group. Dark-phase activity was lowest in late pregnancy, with a decrease in wakefulness. Activity onset after dark-phase onset was delayed in pregnant compared with non-pregnant mice. These behavioural changes contribute to positive energy balance during pregnancy.

## Key points

- Circadian rhythms synchronise daily behaviours including eating, drinking and sleep, but how these change in pregnancy is unclear.
- Food intake increased, with delays in peaks of food intake behaviour late in the light phase from days 6.5 to 12.5 of pregnancy, in comparison to the non-pregnant group.
- The onset of activity after lights off (dark phase) was delayed in pregnant compared with non-pregnant mice.
- Activity decreased by ∼70% in the pregnant group, particularly in the dark (active) phase, with delays in peaks of wakefulness evident from days 0.5–6.5 of pregnancy onwards.
- These behavioural changes contribute to positive energy balance during pregnancy.
- Delays in circadian behaviours during mouse pregnancy were time period and pregnancy stage specific, implying different regulatory mechanisms.

## Introduction

Pregnancy demands an increased energy supply to support fetal and placental development, deposit energy reserves for lactation and support maternal physiological adaptations (Clarke et al., 2021). Dietary intake increases by ∼10% in the third trimester in women and by ∼20–30% in late pregnancy in mice and rats (reviewed by Clarke et al., 2021). Physical activity tends to be lower in the third trimester of human pregnancy (Most et al., 2019) and is dramatically reduced across the entire duration of pregnancy in mice (Ladyman et al., 2018), also increasing the availability of energy. An important component of energy balance regulation is its strong circadian rhythmicity, enabling the coordination of behaviours including food intake, energy expenditure and sleep.

The circadian system is composed of a series of circadian clocks that exist as a hierarchy. The suprachiasmatic nucleus of the anterior hypothalamus is

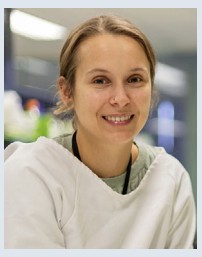

**Georgia S. Clarke** is a recent PhD graduate in the School of Biomedicine at the University of Adelaide. She is interested in pregnancy, nutrition and neuroscience, all of which were encapsulated within her PhD project focused on understanding adaptations in gastrointestinal satiety signalling during pregnancy and the role of pregnancy hormones in driving these changes.

entrained by light and acts as the 'master' clock to entrain other tissues, whilst beneath this, peripheral tissues, including the gastrointestinal tract, contain clock mechanisms to maintain local rhythmicity (Page et al., 2020). These molecular clocks generate circadian rhythms, enabling the daily repeating or synchronisation of events in response to the light–dark cycle and to feeding activity during the 'active' period (Page et al., 2020). These events ultimately regulate physiological functions, optimising energy homeostasis relative to the current environmental demand. Circadian rhythms are important during pregnancy, with disruption of these rhythms increasing the risk of miscarriage, preterm birth and intrauterine growth restriction in women (Cai et al., 2019) and reducing implantation rates and impairing placental development and fetal growth in rodents (Varcoe et al., 2016). Disruption of circadian rhythms during pregnancy also adversely impacts the health of progeny, with impaired neurobehavioural and cognitive outcomes and poorer metabolic health reported in animal models of maternal circadian disruption (Varcoe et al., 2018).

Although the importance of circadian rhythms for pregnancy is appreciated, how pregnancy itself affects circadian rhythms of behaviour and physiological functions is unclear, owing to limited and contradictory evidence. For example, Martin-Fairey et al. (2019) reported an advancement of ≤4 h in the onset of wheel-running activity between gestational day (GD)3 and GD10 in pregnant compared with non-pregnant mice, and Yaw et al. (2021) reported a delayed onset of wheel-running activity between GD8 and GD13. In both studies, the timing of wheel-running activity returned to pre-pregnancy patterns by late pregnancy (Martin-Fairey et al., 2019; Yaw et al., 2021). Consistent with their results in mice, Martin-Fairey et al. (2019) reported that sleep onset advanced by 24 min in the first trimester and by 18 min in the second trimester of human pregnancy. Altered circadian regulation is also evident in the observed changes in feeding behaviours during pregnancy. We have reported that mid- and late-pregnant mice eat more than non-pregnant mice during the light phase, in association with increased meal size (Li et al., 2021). In rats, food and water intake during the dark phase is greater in pregnant than in non-pregnant females (Neubauer & Mletzko, 1990). During rat pregnancy, maximum food intake occurs within a shorter feeding window [between 11 and 15 h after lights on, or Zeitgeber (ZT)11–15 cf. ZT11–19] and follows a bimodal rather than unimodal pattern (Neubauer & Mletzko, 1990). However, circadian patterns of sleep–wake and water intake behaviours have not been reported in mice. Furthermore, it is unknown whether the circadian window of feeding is altered during pregnancy in mice. To address some of these gaps in knowledge, we assessed the circadian rhythm of food and water intake, activity and wakefulness across healthy pregnancy compared with non-pregnant control mice.

## Methods

### Ethical approval

All experimental procedures were approved by the South Australian Health and Medical Research Institute (SAHMRI) Animal Ethics Committee (SAM395.19) and were conducted in compliance with the Australian code for the care and use of animals for scientific purposes, 8th edition 2013. We also confirm that we understand *The Journal of Physiology*'s ethical principles and we comply with the checklists given to authors (du Sert et al., 2018; Grundy, 2015).

### Animals and experimental design

Housing, nutrition and mating of mice has been described previously (Li et al., 2021). Briefly, adult female C57BL/6 mice (10–12 weeks, 18–22 g) were exposed to a 12 h–12 h light–dark cycle (lights on at 07.00 h; ZT0) and fed a standard chow diet *ad libitum*. All mice were housed singly in metabolic cages (Promethion Sable System; Las Vegas, NV, USA) equipped with a food and water hopper/scale system but no running wheel, and acclimated for 7 days. After this period, mice were randomised using a simple table method to be either mated with a stud male to generate pregnancies ($n = 31$) or unmated for study as age-matched, non-pregnant controls ($n = 12$). Mice with vaginal plugs indicative of mating and non-pregnant mice were then placed back into metabolic cages, and data were collected until the mice were terminated at various time points for use in a previously published study of gastric vagal afferent function (Li et al., 2021). Pregnant mice were anaesthetised by isoflurane inhalation (at 5% in oxygen) before humane killing via decapitation in early pregnancy (6.5 days after mating, $n = 10$), mid-pregnancy (12.5 days after mating, $n = 10$) or late pregnancy (17.5 days after mating, $n = 11$). Pregnant mice were randomised using a block method, avoiding weekends and with no more than two mice killed each day, to permit electrophysiological studies previously reported (Li et al., 2021). Non-pregnant mice were killed on age-matched days ($n = 4$ early, $n = 2$ mid, $n = 6$ late) and randomised using the same process. The mice were monitored daily and displayed a behavioural phenotype consistent with reports in other healthy pregnancy studies. This includes a significant increase in maternal body weight by day 7, increases in food intake during mid-pregnancy, primarily attributable to meal size and duration rather than the number of meals (Ladyman

et al., 2018; Li et al., 2021), and a dramatic reduction in physical activity after mating (Ladyman et al., 2018). The number of fetuses were counted in all pregnancies at termination to ensure that the number of fetuses was within the expected range. For analysis, behavioural data were used for all available non-pregnant or pregnant mice on each day after the start of the study.

Mice that were mated and showed vaginal plugs but did not become pregnant ($n = 6$) were excluded from the study and not included in the final group numbers. These mice were not added to the non-pregnant group, in order to avoid potential impacts of elevated prolactin during pseudo-pregnancy (Phillipps et al., 2020). One non-pregnant mouse was excluded owing to an infection at the time of tissue collection and was not included in the final number in the non-pregnant group. The sample size was calculated based on variation in gastric vagal afferent function, which was the primary outcome in the previous study, in which we collected the detailed data on behaviours of these mice (Li et al., 2021). G.S.C. and A.D.V. were not blinded to the groups.

## Metabolic monitoring and data preparation

The following outcomes were recorded by the metabolic cage system: food intake (reduction in hopper weight, intake of <0.002 g excluded), water consumption (reduction in hopper weight), activity (sum of all distances and including fine movement, such as grooming and scratching) and sleep (defined as stillness lasting ≥40 s, converted to wakefulness and modelled as the percentage of time spent sleeping). Metabolic data were recorded continuously at 1 s intervals throughout the study. Raw data were collected by Sablescreens (Promethion Sable System) and extracted using ExpeData v.1.9314 (Promethion Sable System) and Macro Interpreter v.2.44 (Promethion Sable System). Food and water intake and activity data were analysed using the Universal Macro Collection v.10.1.1.2 (macro 13), and sleep data were analysed using OneClickMacro 5 min intervals v.2.50.4.4 (macro 13). These macros provided food and water intake, activity and wake data for each individual mouse averaged across 5 min blocks. Data were then averaged within each mouse for each hour of study before circadian analysis. Data for each outcome were analysed in three time blocks, corresponding to thirds of mouse pregnancy in the pregnant group: days 0.5–6.5, 6.5–12.5 and 12.5–17.5 of the study. These time blocks represent stages of developmental progression including implantation of the blastocyst and placental development during days 5–8, with growth of the blastocyst implantation site; definite placenta structure present by days 10 and 11; and the placenta at maximum size during days 15–17 (Panja & Paria, 2021).

Unlike previous studies, in which analyses of activity have been based on movement of a running wheel (Martin-Fairey et al., 2019; Yaw et al., 2021), we recorded spontaneous activity within the cage. We therefore developed criteria to define the onset of activity based on cage activity measures. We first defined a threshold activity for each individual mouse as the average of dark-phase activity during the acclimation period. We defined the onset of dark-phase activity for each study day as the first time at which activity exceeded this threshold for the next three consecutive 5 min blocks. Confirming that this definition captured normal dark-phase onset behaviour in non-pregnant mice, activity onset occurred within 2 h of lights off for 95.2% of data sets collected during the acclimation period, including two full dark-phase cycles, and all mice achieved activity onset within 2 h of lights off on at least one acclimation night. For mice that did not achieve activity onset within the dark phase of any study day, activity onset time was set at 12 h for analysis. Overall, there were 12, 8 and 6 non-pregnant and 31, 21 and 11 pregnant mice with data for study days 0.5–6.5, 6.5–12.5 and 12.5–17.5, respectively. All data points were used for each animal.

## Statistical methods

Raw data for food and water intake, activity and time spent awake are presented as the mean (SD) for each mouse, averaged across each study block, day 0.5–6.5, 6.5–12.5 and 12.5–17.5, within non-pregnant and pregnant groups. Owing to data distributions, we modelled the fraction of time spent awake/asleep (ranging from zero to one) with a $\beta$ distribution. For the other three outcomes (range ≥ 0), analyses on the original scale produced residual distributions that were clearly not normally distributed; a square root transformation was used to resolve this problem. Square-rooted outcomes were back-transformed as the square of the square root mean plus the square root variance. To capture the rapid changes in mean levels over the time period, natural splines were used, with 13 knots evenly spaced from ZT8.5 to ZT4.5.

A two-stage approach was used to determine effects of pregnancy within each study day block, day 0.5–6.5, 6.5–12.5 and 12.5–17.5. To compare differences in peak time and location between treatment groups (pregnancy *vs.* non-pregnancy) requires a complex model that allows rapid variation in behaviour and peak location. The traditional two-way ANOVA (time × group) does not allow for assessment of the timing of peak behaviour. Therefore, we explored models of behaviour change that allowed activity to vary continuously over time using splines and aimed to capture the variation in behaviour observed in the raw data. This model is a not mechanistic model, but an exploratory model, and it allowed us to

identify differences in behaviour between groups and over time [days since pregnant (DSP)]. Initially, we explored three-way interaction models of time (spline, hours) × group × linear DSP. However, it became apparent that non-linear modelling for the third component (DSP) would be required. The three-way interaction: spline (time, hours) × group × spline DSP had too many parameters, and simpler models using polynomials for the third term resulted in poor estimation at the extremes (days 0 and 17). Hence, we present the simpler model with DSP discretised into three categories.

The acclimation period was modelled to generate within-individual estimates (on the logit scale for the wake–sleep model and square-root scale for activity, food and water intake models) for each mouse. These models included the natural splines (13 knots) with interaction with pregnancy group, a random intercept per mouse, and were estimated using the R package *glmmTMB* (Brooks et al., 2017).

Similar multi-level models were constructed in order to compare pregnant and non-pregnant mice. These models included the same natural splines (13 knots) with a three-way interaction with pregnancy status and study day as fixed effects and nested random intercepts per study day block within mouse. Individual variation in behaviour was corrected for by inclusion of the within-individual acclimation period estimates of each behaviour across time. Bayesian STAN (Stan Development Team, 2023b) model code was generated using the package *brms* (Bürkner, 2017) and estimated using the package *rstan* (Stan Development Team, 2023a). The non-informative priors recommended by *brms* were used. Each model was estimated using three chains with 10,000 iterations, half used for burn-in and thinned by a quarter. We note that the differences in the timing of peak activity could have been analysed using bootstrapped multilevel mixed-effects models; however, model convergence is often a problem for inference via likelihood, hence our choice to use the Bayesian Monte Carlo methodology.

Visual inspection of the raw and modelled data identified five time periods when differing local behaviour occurred: period I, ZT8–ZT12 (late light-phase); period II, ZT12–ZT15 (early dark-phase); period III, ZT15–ZT18 (mid dark-phase); period IV, ZT18–ZT24 (late dark-phase); and period V, ZT24–ZT4 (early light-phase). For each behavioural outcome within each time period, we assessed whether a peak in behaviour occurred (local maximum), and peak time and amplitude were extracted for detected peaks. Mean and 2.5 and 97.5 quantiles across the Bayesian model iterations are reported for point and interval estimates for each group (non-pregnant and pregnant) and by study day block, day 0.5–6.5, 6.5–12.5 and 12.5–17.5. Peak time and location were considered statistically significant if the credible interval did not cross over zero. The raw data and codes

for this model have been made available on figshare (https://doi.org/10.25909/c.7111231.v1)

Timing of activity onset was analysed using a mixed model, with pregnancy status (between-animal factor) and study day (within-animal factor), using SPSS v.28 (IBM Corporation, Armonk, NY, USA).

## Results

### Effects of pregnancy on food intake

Modelled food intake patterns corresponded to raw data in each of the three blocks of the study (Fig. 1*A*–*C*). Modelled food intake was similar in non-pregnant and pregnant mice at all time points on days 0.5–6.5 of the study (Fig. 1*D*). During days 6.5–12.5 of study, modelled food intake was greater in pregnant than in non-pregnant mice only between ZT10 and ZT11 (Fig. 1*E*). In the final block, day 12.5–17.5, modelled food intake was greater in pregnant than non-pregnant mice between ZT10 and ZT11 and also between ZT13 and ZT14 (Fig. 1*F*).

**Days 0.5–6.5.** Within the first time period of interest (ZT8–ZT12), 57% of non-pregnant and 33% of the pregnant group (model iterations) exhibited a peak in food intake (Table 1). The timing and amplitude of peaks in food intake, for those animals that exhibited a peak, was similar in the non-pregnant and pregnant groups (Table 1). In the second time period of interest (ZT12–ZT15), 65% of the non-pregnant group and 99% of the pregnant group exhibited a peak (Table 1). Within this time period, food intake was 1.3-fold higher in the pregnant than the non-pregnant group, but peak timing was similar between groups. In later time periods (ZT15–ZT18, ZT18–ZT24 and ZT24–ZT4), 89, 99 and 99% of the non-pregnant group and 25, 100 and 99% of the pregnant group, respectively, exhibited peaks, and the peak timing and amplitude did not differ between non-pregnant and pregnant groups (Table 1).

**Days 6.5–12.5.** Within the first time period of interest (ZT8–ZT12), 85% of the non-pregnant group and 36% of the pregnant group exhibited peaks in food intake (Table 1). Food intake in the pregnant group was 2.0-fold greater and the timing was 1.62 h later than in the non-pregnant group. In the second time period of interest (ZT12–ZT15), 72% of the non-pregnant group and 100% of the pregnant group exhibited peaks, and again food intake was 1.4-fold greater in the pregnant than the non-pregnant group, although peak timing was similar between groups (Table 1). In the third, fourth and fifth time periods of interest (ZT15–ZT18, ZT18–ZT24 and ZT24–ZT4), 74, 99 and 100% of the non-pregnant group

**Table 1. The effect of pregnancy on timing and amplitudes of peaks in food intake**

| Time period of interest | Study block | Iterations per group with detected peak (%) | | Peak characteristics for mice with detected peak | | | | | |
|---|---|---|---|---|---|---|---|---|---|
| | | | | Peak time (ZT) [95% CrI] | | | Peak amplitude (g/h) [95% CrI] | | |
| | | Non-pregnant | Pregnant | Non-pregnant | Pregnant | Difference | Non-pregnant | Pregnant | Difference |
| I (ZT8–ZT12) | 1 | 57.1 | 33.1 | 9.22 [8.79, 9.81] | 9.18 [8.52, 9.58] | −0.05 [−0.90, 0.58] | 0.083 [0.057, 0.115] | 0.074 [0.058, 0.092] | −0.009 [−0.044, 0.023] |
| | 2 | 85.5 | 36.0 | 9.12 [8.58, 9.87] | 10.74 [10.47, 11.05] | **1.62 [0.70, 2.22]** | 0.077 [0.046, 0.115] | 0.156 [0.126, 0.189] | **0.079 [0.030, 0.124]** |
| | 3 | 66.1 | 57.2 | 9.20 [8.30, 10.73] | 10.79 [10.52, 11.16] | **1.60 [0.07, 2.47]** | 0.067 [0.036, 0.106] | 0.205 [0.159, 0.256] | **0.138 [0.077, 0.197]** |
| II (ZT12–ZT15) | 1 | 65.1 | 99.1 | 13.31 [12.27, 14.64] | 13.74 [13.41, 14.29] | 0.43 [−0.94, 1.62] | 0.212 [0.169, 0.260] | 0.270 [0.234, 0.307] | **0.058 [0.000, 0.114]** |
| | 2 | 71.9 | 100 | 13.89 [12.74, 14.79] | 13.83 [13.65, 14.11] | −0.06 [−0.97, 1.11] | 0.273 [0.212, 0.337] | 0.373 [0.325, 0.0423] | **0.101 [0.018, 0.179]** |
| | 3 | 48.1 | 99.3 | 13.60 [12.23, 14.94] | 13.93 [13.59, 14.67] | 0.32 [−1.13, 1.88] | 0.214 [0.152, 0.283] | 0.365 [0.305, 0.431] | **0.152 [0.058, 0.244]** |
| III (ZT15–ZT18) | 1 | 89.0 | 25.4 | 16.38 [15.21, 17.48] | 16.88 [15.32, 17.44] | 0.50 [−1.16, 1.87] | 0.238 [0.201, 0.278] | 0.213 [0.190, 0.236] | −0.026 [−0.071, 0.017] |
| | 2 | 73.7 | 35.3 | 16.64 [15.15, 17.69] | 16.98 [16.46, 17.40] | 0.34 [−0.90, 1.91] | 0.263 [0.218, 0.0311] | 0.274 [0.244, 0.306] | 0.011 [−0.047, 0.067] |
| | 3 | 80.6 | 5.3 | 15.99 [15.10, 17.41] | 16.80 [15.04, 17.58] | 0.81 [−1.22, 2.19] | 0.254 [0.196, 0.318] | 0.244 [0.205, 0.284] | −0.010 [−0.087, 0.060] |

*(Continued)*

**Table 1. (Continued)**

| Time period of interest | Study block | Iterations per group with detected peak (%) | | Peak characteristics for mice with detected peak | | | | | |
| --- | --- | --- | --- | --- | --- | --- | --- | --- | --- |
| | | | | Peak time (ZT) [95% CrI] | | | Peak amplitude (g/h) [95% CrI] | | |
| | | Non-pregnant | Pregnant | Non-pregnant | Pregnant | Difference | Non-pregnant | Pregnant | Difference |
| IV (ZT18–ZT24) | 1 | 99.3 | 100 | 22.12 [21.76, 22.32] | 22.24 [22.14, 22.33] | 0.11 [−0.11, 0.48] | 0.167 [0.127, 0.208] | 0.213 [0.183, 0.245] | 0.047 [−0.003, 0.096] |
| | 2 | 99.9 | 100 | 22.23 [21.95, 22.41] | 22.35 [22.26, 22.42] | 0.12 [−0.08, 0.40] | 0.242 [0.184, 0.306] | 0.266 [0.255, 0.311] | 0.024 [−0.053, 0.098] |
| | 3 | 100 | 100 | 22.16 [21.95, 22.60] | 22.27 [22.05, 22.42] | 0.10 [−0.15, 0.35] | 0.297 [0.223, 0.379] | 0.225 [0.177, 0.282] | −0.071 [−0.167, 0.023] |
| V (ZT24–ZT4) | 1 | 99.7 | 99.4 | 1.53 [1.31, 1.79] | 1.90 [1.53, 2.87] | 0.37 [−0.12, 1.37] | 0.089 [0.060, 0.123] | 0.073 [0.058, 0.090] | −0.016 [−0.052, 0.017] |
| | 2 | 69.9 | 92.4 | 1.56 [1.03, 3.65] | 2.59 [1.60, 3.73] | 1.02 [−1.08, 2.36] | 0.055 [0.031, 0.086] | 0.060 [0.043, 0.080] | 0.005 [−0.030, 0.037] |
| | 3 | 99.5 | 79.0 | 1.55 [1.33, 1.88] | 2.96 [1.56, 3.91] | 1.41 [−0.02, 2.38] | 0.069 [0.036, 0.112] | 0.081 [0.052, 0.116] | 0.012 [−0.039, 0.060] |

Abbreviations: CrI, 95% credible interval; study block = block 1 (days 0.5–6.5, *n* = 31), block 2 (days 6.5–12.5, *n* = 21), block 3 (days 12.5–17.5, *n* = 11); non-pregnant mice, *n* ≥ 6; ZT, Zeitgeber. Intervals of group differences that exclude zero are shown in bold text. Positive values present delayed timing or increased amplitude. Negative values present advanced timing or decreased amplitude.

and 35, 100 and 92% of the pregnant group, respectively, exhibited peaks, and there was no difference in peak timing or amplitude between groups (Table 1).

**Days 12.5–17.5.** Within the first time period of interest (ZT8–ZT12), 66% of the non-pregnant group and 57% of the pregnant group exhibited a peak in food intake (Table 1). Within the mice that exhibited this peak, food intake was 3.10-fold greater and the timing was 1.60 h later in pregnant than non-pregnant mice. In the second time period of interest (ZT12–ZT15), 48% of non-pregnant and 99% of the pregnant group exhibited peaks (Table 1). Food intake at this peak was 1.7-fold greater during pregnancy, whilst peak timing was similar in the pregnant and non-pregnant groups. In the third, fourth and fifth time periods of interest (ZT15–ZT18, ZT18–ZT24 and ZT24–ZT4), 80, 100 and 95% of the non-pregnant group and 5, 100 and 79% of the pregnant group, respectively, exhibited peaks (Table 1). For each time period, within the mice with peaks, there were no differences between the pregnant and non-pregnant groups in the timing or amplitude of these peaks (Table 1).

### Effects of pregnancy on water intake

Modelled water intake patterns corresponded to raw data in each three blocks of the study (Fig. 2*A–C*). Modelled water intake was similar in non-pregnant and pregnant mice at all time points during days 0.5–6.5 and 6.5–12.5 of the study (Fig. 2*D* and *E*). During days 12.5–17.5, modelled water intake was greater in pregnant than non-pregnant mice between ZT13 and ZT14 (Fig. 2*F*).

**Days 0.5–6.5.** Within the first, second, fourth and fifth time periods of interest (ZT8–ZT12, ZT12–ZT15, ZT18–ZT24 and ZT24–ZT4) 64, 57, 100 and 100% of the non-pregnant group and 26, 99, 100 and 100% of pregnant group, respectively, exhibited peaks in water intake (Table 2). Within mice with a peak in these time periods, there were no differences in the timing or amplitude of water intake peaks between the pregnant and non-pregnant groups (Table 2). In the third time period of interest (ZT15–ZT18), 98% of the non-pregnant group and 92% of the pregnant group exhibited a peak in water intake, with similar timing in both groups and with a 0.2-fold lower peak amplitude in the pregnant group than in the non-pregnant group (Table 2).

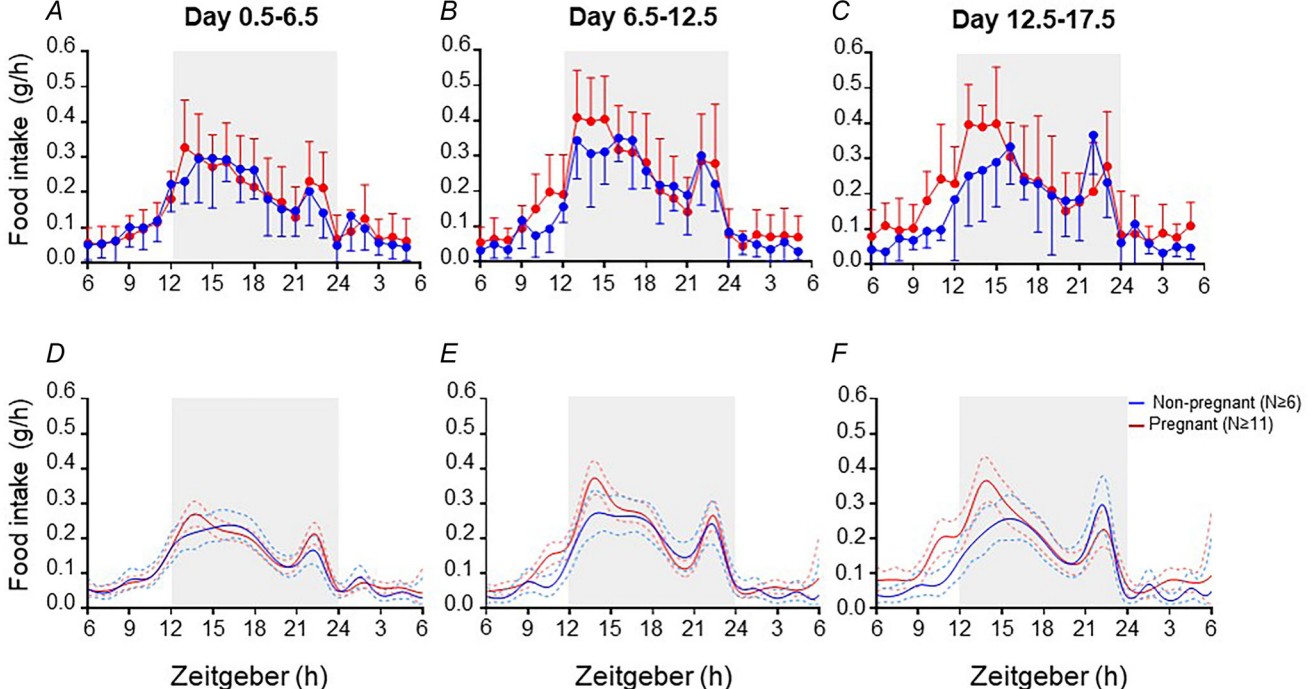

**Figure 1. Food intake pattern of non-pregnant and pregnant mice**
Food consumption (*A–C*) is the mean (SD) of raw data for each mouse, averaged across each study block (day 0.5–6.5, 6.5–12.5 and 12.5–17.5), within non-pregnant (blue line, *n* ≥ 6) and pregnant (red line, *n* ≥ 11) groups. The light phase (ZT6–ZT12 and ZT24–ZT6, no shading) and dark phase (ZT12–ZT24, shaded) are shown for day 0.5–6.5 (*A*), day 6.5–12.5 (*B*) and day 12.5–17.5 (*C*). The fitted model of food consumption (*D–F*) across each of the three blocks of the study indicates fitted means (continuous lines) and 95% credible intervals (dashed lines) for all non-pregnant (blue) and pregnant (red) groups. The light phase (ZT6–ZT12 and ZT24–ZT6, no shading) and dark phase (ZT12–ZT24, shaded) are shown for day 0.5–6.5 (*D*), day 6.5–12.5 (*E*) and day 12.5–17.5 (*F*).

**Table 2. The effect of pregnancy on timing and amplitudes of peaks in water intake**

| Time period of interest | Study block | Iterations per group with detected peak (%) | | Peak characteristics for mice with detected peak | | | | | |
|---|---|---|---|---|---|---|---|---|---|
| | | | | Peak time (ZT) [95% CrI] | | | Peak amplitude (g/h) [95% CrI] | | |
| | | Non-pregnant | Pregnant | Non-pregnant | Pregnant | Difference | Non-pregnant | Pregnant | Difference |
| I (ZT8–ZT12) | 1 | 64.4 | 26.4 | 8.77 [8.07, 9.29] | 9.08 [8.15, 9.55] | 0.30 [−0.53, 1.15] | 0.071 [0.048, 0.097] | 0.060 [0.046, 0.076] | −0.011 [−0.039, 0.016] |
| | 2 | 72.6 | 60.5 | 9.06 [8.62, 9.49] | 10.42 [9.27, 10.89] | **1.36 [0.15, 2.04]** | 0.076 [0.047, 0.011] | 0.106 [0.081, 0.132] | 0.030 [−0.014, 0.071] |
| | 3 | 39.2 | 86.5 | 9.24 [8.22, 11.05] | 10.40 [9.31. 10.80] | 1.16 [−0.68, 2.30] | 0.074 [0.042, 0.011] | 0.167 [0.125, 0.215] | **0.094 [0.039, 0.150]** |
| II (ZT12–ZT15) | 1 | 57.9 | 99.9 | 13.62 [12.56, 14.40] | 13.71 [13.55, 13.91] | 0.09 [−0.71, 1.16] | 0.027 [0.018, 0.028] | 0.256 [0.224, 0.289] | 0.029 [−0.029, 0.085] |
| | 2 | 69.2 | 99.9 | 13.19 [12.34, 14.21] | 13.90 [13.71, 14.22] | 0.71 [−0.37, 1.61] | 0.236 [0.186, 0.291] | 0.280 [0.241, 0.320] | 0.043 [−0.022, 0.107] |
| | 3 | 43.3 | 100 | 13.27 [12.14, 14.79] | 13.71 [13.60, 13.83] | 0.44 [−1.07, 1.58] | 0.220 [0.164, 0.284] | 0.454 [0.383, 0.526] | **0.234 [0.140, 0.331]** |
| III (ZT15–ZT18) | 1 | 98.4 | 92.4 | 16.90 [15.82, 17.56] | 17.23 [16.61, 17.60] | 0.32 [−0.59, 1.51] | 0.262 [0.226, 0.301] | 0.213 [0.190, 0.236] | **−0.049 [−0.095, −0.006]** |
| | 2 | 98.2 | 47 | 16.86 [15.67, 17.61] | 17.06 [16.45, 17.52] | 0.20 [−0.82, 1.48] | 0.284 [0.239, 0.333] | 0.218 [0.191, 0.246] | **−0.067 [−0.124, −0.012]** |
| | 3 | 91.7 | 73.2 | 16.28 [15.20, 17.61] | 17.53 [17.01, 17.89] | 1.25 [−0.18, 2.39] | 0.286 [0.232, 0.345] | 0.254 [0.214, 0.297] | −0.032 [−0.106, 0.039] |

*(Continued)*

**Table 2. (Continued)**

| Time period of interest | Study block | Iterations per group with detected peak (%) | | Peak characteristics for mice with detected peak | | | | | |
| --- | --- | --- | --- | --- | --- | --- | --- | --- | --- |
| | | | | Peak time (ZT) [95% CrI] | | | Peak amplitude (g/h) [95% CrI] | | |
| | | Non-pregnant | Pregnant | Non-pregnant | Pregnant | Difference | Non-pregnant | Pregnant | Difference |
| IV (ZT18–ZT24) | 1 | 100 | 100 | 22.24 [22.13, 22.33] | 22.31 [22.24, 22.36] | 0.06 [−0.05, 0.19] | 0.267 [0.217, 0.320] | 0.240 [0.210, 0.273] | −0.028 [−0.088, 0.032] |
| | 2 | 100 | 100 | 22.31 [22.17, 22.42] | 22.35 [22.27, 22.42] | 0.04 [−0.10, 0.20] | 0.309 [0.246, 0.377] | 0.260 [0.222, 0.300] | −0.049 [−0.126, 0.024] |
| | 3 | 100 | 100 | 22.14 [21.85, 22.31] | 22.48 [22.31, 22.74] | **0.34 [0.09, 0.72]** | 0.313 [0.242, 0.390] | 0.251 [0.202, 0.303] | −0.062 [−0.155, 0.029] |
| V (ZT24–ZT4) | 1 | 100 | 100 | 1.58 [1.44, 1.76] | 1.90 [1.61, 2.48] | 0.32 [−0.02, 0.92] | 0.124 [0.091, 0.162] | 0.095 [0.077, 0.114] | −0.029 [−0.070, 0.010] |
| | 2 | 96.9 | 84.2 | 2.24 [1.36, 3.94] | 2.99 [1.67, 3.91] | 0.74 [−1.47, 2.26] | 0.086 [0.058, 0.119] | 0.079 [0.058, 0.102] | −0.007 [−0.046, 0.031] |
| | 3 | 97.8 | 85.2 | 1.08 [1.46, 3.42] | 1.42 [1.66, 3.96] | 1.34 [−0.46, 2.31] | 0.097 [0.063, 0.137] | 0.122 [0.090, 0.159] | 0.026 [−0.025, 0.075] |

Abbreviations: CrI, 95% credible interval; study block = block 1 (days 0.5–6.5, *n* = 31), block 2 (days 6.5–12.5, *n* = 21), block 3 (days 12.5–17.5, *n* = 11); non-pregnant mice, *n* ≥ 6; ZT, Zeitgeber. Intervals of group differences that exclude zero are shown in bold text. Positive values present delayed timing or increased amplitude. Negative values present advanced timing or decreased amplitude.

**Days 6.5–12.5.** Within the first time period of interest (ZT8–ZT12), 72% of the non-pregnant group and 60% of the pregnant group exhibited peaks in water intake (Table 2). Within mice exhibiting a peak, the peak occurred 1.36 h later in the pregnant group than in the non-pregnant group, with similar amplitude in both groups (Table 2). In the third time period of interest (ZT15–ZT18), 98% of the non-pregnant group and 47% of the pregnant group exhibited a peak, with a 0.2-fold lower peak amplitude in the pregnant group than in non-pregnant group, and no difference in peak timing (Table 2). In the second, fourth and fifth time periods of interest (ZT12–ZT15, ZT18–ZT24 and ZT24–ZT4), 69, 100 and 97% of the non-pregnant group and 99, 100 and 84% of the pregnant group, respectively, exhibited peaks, with no differences in peak amplitude or timing between the pregnant and non-pregnant groups for these peaks (Table 2).

**Days 12.5–17.5.** Within the first and second time periods of interest (ZT8–ZT12 and ZT12–ZT15), 39 and 43% of the non-pregnant group and 86 and 100% of the pregnant group, respectively, exhibited a peak in water intake (Table 2). Within mice exhibiting these peaks, the amplitude of the water intake peak was 2.3-fold and 2.1-fold greater, respectively, in the pregnant group than in the non-pregnant group, and peak timing was similar between groups (Table 2). In the third and fifth time periods of interest (ZT15–ZT18 and ZT24–ZT4), 91 and 97% of the non-pregnant group and 73 and 85% of the pregnant group, respectively, exhibited peaks (Table 2), and the peak timing and amplitude were similar in the pregnant and non-pregnant groups. In the fourth time period of interest (ZT18–ZT24), all non-pregnant and pregnant groups exhibited a peak (Table 2). This peak in water intake occurred 0.34 h later in the pregnant group than in the non-pregnant group, and peak amplitude was similar between groups.

## Effects of pregnancy on dark-phase activity onset

The timing of activity onset is illustrated in Fig. 3, including examples of activity onset during the acclimation period and on day 8 of the study in non-pregnant (Fig. 3*A* and *B*, respectively) and pregnant (Fig. 3*C* and *D*, respectively) mice. The time after lights off at which mice achieved their individual activity threshold changed in a different way across

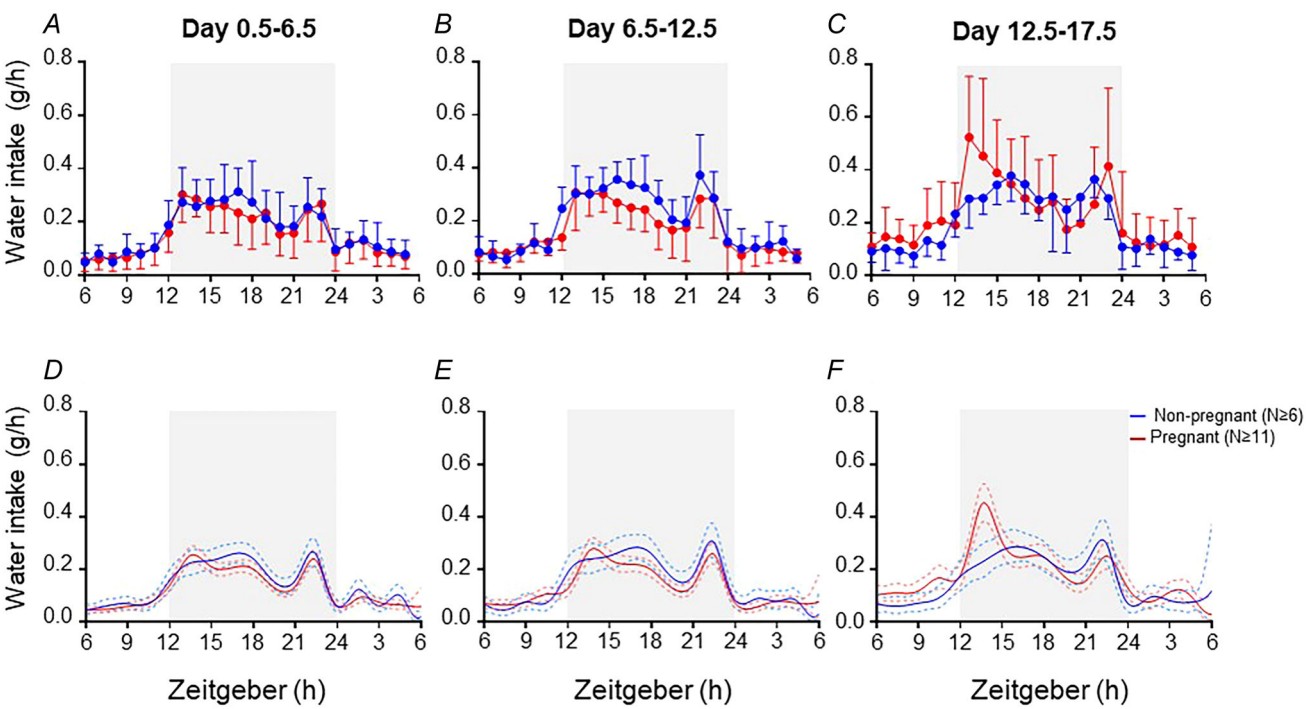

**Figure 2. Water intake pattern of non-pregnant and pregnant mice**
Water consumption (*A–C*) is the mean (SD) of raw data for each mouse, averaged across each study block (day 0.5–6.5, 6.5–12.5 and 12.5–17.5), within non-pregnant (blue line, *n* ≥ 6) and pregnant (red line, *n* ≥ 11) groups. The light phase (ZT6–ZT12 and ZT24–ZT6, no shading) and dark phase (ZT12–ZT24, shaded) are shown for day 0.5–6.5 (*A*), day 6.5–12.5 (*B*) and day 12.5–17.5 (*C*). The fitted model of water consumption (*D–F*) across each of the three blocks of the study indicates fitted means (continuous lines) and 95% credible intervals (dashed lines) for all non-pregnant (blue) and pregnant (red) groups. The light phase (ZT6–ZT12 and ZT24–ZT6, no shading) and dark phase (ZT12–ZT24, shaded) are shown for day 0.5–6.5 (*D*), day 6.5–12.5 (*E*) and day 12.5–17.5 (*F*).

the study in non-pregnant and pregnant mice (Fig. 3E; day × pregnancy interaction, $P = 0.042$). In non-pregnant mice, the timing of activity onset remained consistent across days (Fig. 3E; $P = 0.352$), being similar during acclimation and during the study, as shown for an individual mouse (Fig. 3A and B). In contrast, pregnant mice took longer to reach their activity threshold as pregnancy progressed (Fig. 3E; $P < 0.001$). The example in Fig. 3C and D shows data for an individual mouse that achieved activity onset at 40 min after lights off during an acclimation night but did not achieve its individual activity threshold at day 8 of pregnancy. Remarkably, activity onset was delayed in pregnant mice even at day 1 of pregnancy [difference 1.86 (1.99) h, $P = 0.049$], and

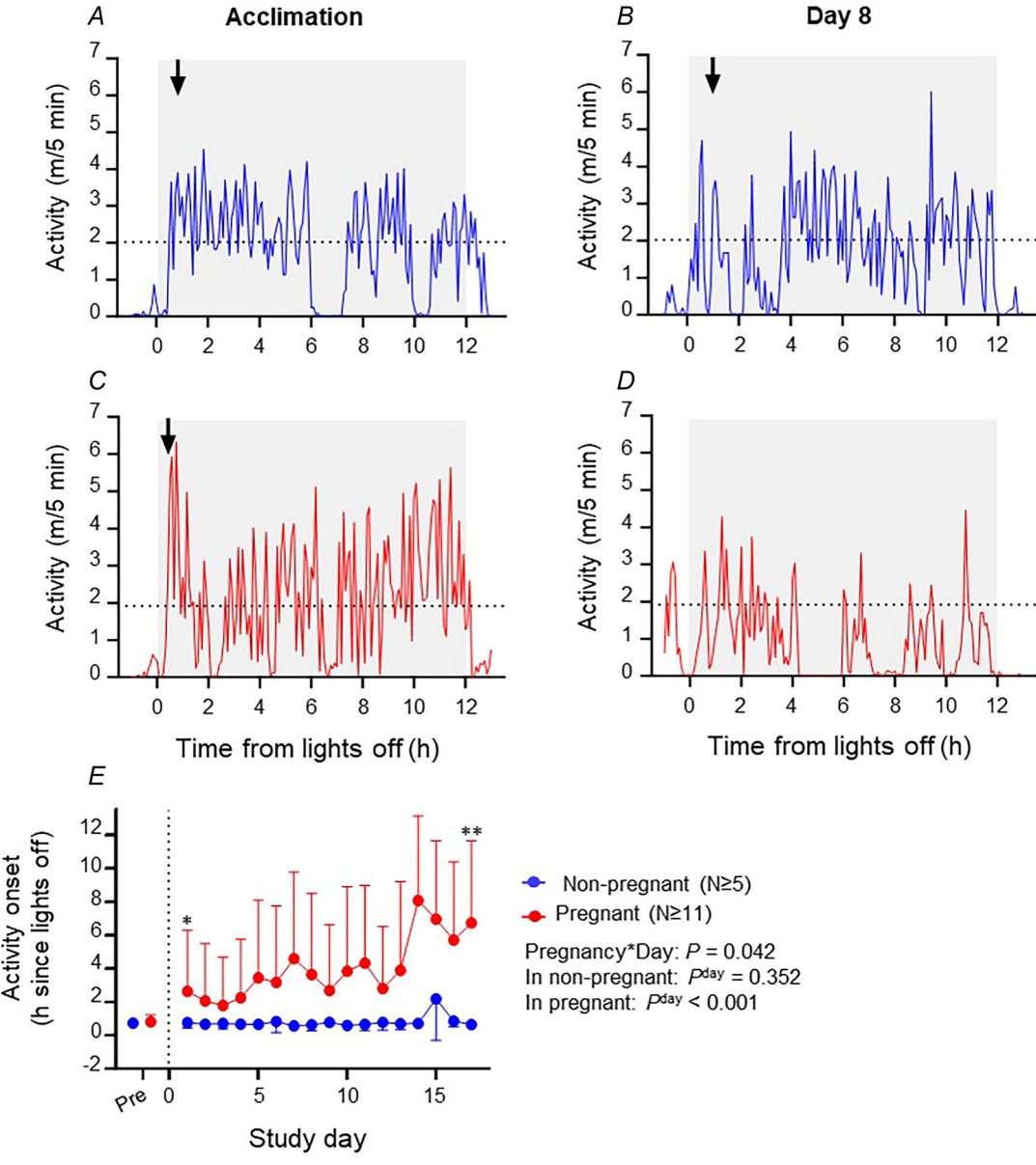

**Figure 3. The impact of pregnancy on the time of first activity onset during the dark phase**
Daily activity onset in the dark phase was defined as the time at which the mouse first achieved three consecutive 5 min periods of movement greater than its the average night-time activity during the acclimation period. Representative plots of cage activity recorded every 5 min are shown for acclimation (A and C) and study day 8 (B and D) for a representative non-pregnant (A and B) and pregnant (C and D) mouse during the dark phase (shaded area). The dotted line indicates the acclimation average dark-phase activity for the individual mouse, and the downwards arrow shows the time at which the mouse met the criterion for activity onset. The pregnant mouse on day 8 (D) did not reach the criterion for activity onset. Average timing of dark-phase activity onset in non-pregnant (blue symbols and line, $n \geq 5$) and pregnant (red symbols and line, $n \geq 11$) mice is presented as the mean (SD) for the acclimation period and for each study day (E). *$P < 0.05$, *$P < 0.01$ v. non-prengnant mice.

at day 17 of the study the activity onset was 6.07 (5.17) h later in pregnant than non-pregnant mice ($P = 0.007$).

### Effect of pregnancy on activity

Modelled total activity patterns corresponded to raw data in each three blocks of the study (Fig. 4*A*–*C*). For all days of the study, modelled total activity was similar in non-pregnant and pregnant mice throughout the light period (ZT0–ZT12), and activity throughout the majority of the dark period (ZT13–ZT23) was lower in pregnant compared with non-pregnant mice (Fig. 4*D*–*F*).

**Days 0.5–6.5.** Within all time periods of interest (ZT8–ZT12, ZT12–ZT15, ZT15–ZT18, ZT18–ZT24 and ZT24–ZT4), >94% of the non-pregnant group and >86% of the pregnant group exhibited a peak in activity (Table 3). Within mice that exhibited a peak in activity, the peak amplitude was lower for all time periods (0.4-, 0.4-, 0.7-, 0.5- and 0.4-fold, respectively) in the pregnant group than in the non-pregnant group (Table 3). In the fourth time period of interest (ZT18–ZT24), the timing of the peak occurred 0.20 h later in the pregnant than in the

non-pregnant group, but peaks occurred at similar times for all other periods of interest (Table 3).

**Days 6.5–12.5.** Within the first and fifth time periods of interest (ZT8–ZT12 and ZT24–ZT4), 73 and 66% of the non-pregnant group and 34 and 83% of the pregnant group, respectively, exhibited peaks in activity, and detected peaks were of similar timing and amplitudes in both groups (Table 3). In the second, third and fourth time periods of interest (ZT12–ZT15, ZT15–ZT18 and ZT18–ZT24), 96, 99 and 100% of the non-pregnant group and 100, 53 and 100% of the pregnant group, respectively, exhibited activity peaks (Table 3). The amplitude of peak activity was lower (0.5-, 0.7- and 0.5-fold, respectively) in the pregnant than in the non-pregnant group for all these time periods during the dark period (Table 3). The activity peak in time period four (ZT18–ZT24) occurred 0.13 h later in the pregnant group than in the non-pregnant group, whilst the timing of activity peaks between ZT12–ZT15 and ZT15–ZT18 were similar in the pregnant and non-pregnant groups (Table 3).

**Days 12.5–17.5.** Within the first time period of interest (ZT8–ZT12), 87% of the non-pregnant group and 41% of

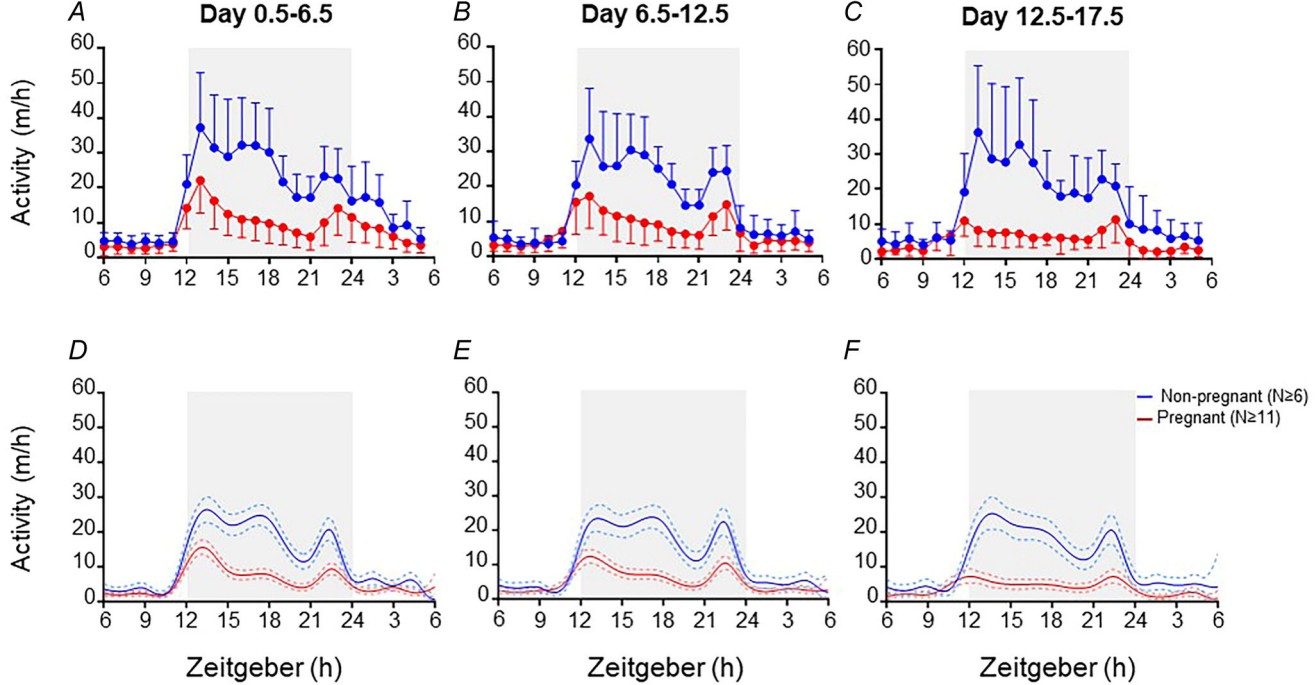

**Figure 4. Activity pattern of non-pregnant and pregnant mice**
Activity (*A*–*C*) is the mean (SD) of raw data for each mouse, averaged across each study block (day 0.5–6.5, 6.5–12.5 and 12.5–17.5), within non-pregnant (blue line, $n \geq 6$) and pregnant (red line, $n \geq 11$) groups. The light phase (ZT6–ZT12 and ZT24–ZT6, no shading) and dark phase (ZT12–ZT24, shaded) are shown for day 0.5–6.5 (*A*), day 6.5–12.5 (*B*) and day 12.5–17.5 (*C*). The fitted model of activity (*D*–*F*) across each of the three blocks of the study indicates fitted means (continuous lines) and 95% credible intervals (dashed lines) for all non-pregnant (blue) and pregnant (red) groups. The light phase (ZT6–ZT12 and ZT24–ZT6, no shading) and dark phase (ZT12–ZT24, shaded) are shown for day 0.5–6.5 (*D*), day 6.5–12.5 (*E*) and day 12.5–17.5 (*F*).

**Table 3. The effect of pregnancy on timing and amplitudes of peaks in activity**

| Time period of interest | Study block | Iterations per group with detected peak (%) | | Peak characteristics for mice with detected peak | | | | | |
| --- | --- | --- | --- | --- | --- | --- | --- | --- | --- |
| | | | | Peak time (ZT) [95% CrI] | | | Peak amplitude (m/h) [95% CrI] | | |
| | | Non-pregnant | Pregnant | Non-pregnant | Pregnant | Difference | Non-pregnant | Pregnant | Difference |
| I (ZT8–ZT12) | 1 | 94.8 | 86.2 | 8.66 [8.17, 8.91] | 8.62 [8.12, 8.93] | −0.03 [−0.62, 0.53] | 3.95 [2.72, 5.36] | 2.30 [1.60, 3.11] | **−1.65 [−3.25, −0.12]** |
| | 2 | 73.5 | 34.9 | 8.60 [8.09, 8.92] | 9.02 [8.38, 9.35] | 0.42 [−0.24, 1.03] | 3.58 [2.30, 5.18] | 2.48 [1.65, 3.48] | −1.10 [−2.94, 0.54] |
| | 3 | 87.4 | 41.5 | 8.80 [8.20, 9.14] | 11.56 [9.88, 11.98] | 2.75 [−0.11, 3.60] | 4.37 [2.68, 6.44] | 6.90 [5.13, 8.86] | 2.53 [−0.20, 5.22] |
| II (ZT12–ZT15) | 1 | 99.8 | 100 | 13.53 [13.12, 13.75] | 13.17 [12.93, 1.35] | −0.30 [−0.67, 0.10] | 26.4 [22.7, 30.1] | 15.6 [13.5, 17.7] | **−10.8 [−15.2, −6.51]** |
| | 2 | 96.9 | 100 | 13.23 [12.68, 13.74] | 12.70 [12.40, 13.11] | −0.53 [−1.17 0.16] | 23.4 [19.7, 27.3] | 12.4 [10.5, 14.4] | **−11.0 [−15.5, −6.67]** |
| | 3 | 95.8 | 62.6 | 13.65 [12.94, 13.54] | 12.45 [12.01, 13.61] | −1.21 [−2.14, 0.16] | 25.3 [20.7, 30.1] | 7.05 [5.29, 9.0] | **−18.2 [−23.3, −13.2]** |
| III (ZT15–ZT18) | 1 | 99.4 | 92.7 | 17.29 [16.71, 17.60] | 17.42 [16.89, 17.70] | 0.13 [−0.42, 0.78] | 24.5 [21.9, 27.8] | 8.00 [6.69, 9.42] | **−16.7 [−20.1, −13.5]** |
| | 2 | 99.0 | 53.6 | 17.10 [16.20, 17.55] | 17.09 [16.45, 17.54] | −0.01 [−0.86, 1.03] | 23.8 [20.6, 27.1] | 7.12 [5.82, 8.56] | **−16.7 [−20.4, −13.2]** |
| | 3 | 47.3 | 76.4 | 16.83 [15.17, 17.61] | 16.94 [15.19, 17.92] | 0.13 [−1.95, 2.23] | 21.0 [17.7, 24.5] | 4.96 [3.73, 6.36] | **−16.0 [−19.7, −12.4]** |

*(Continued)*

**Table 3. (Continued)**

| Time period of interest | Study block | Iterations per group with detected peak (%) | | Peak characteristics for mice with detected peak | | | | | |
| | | | | Peak time (ZT) [95% CrI] | | | Peak amplitude (m/h) [95% CrI] | | |
| | | Non-pregnant | Pregnant | Non-pregnant | Pregnant | Difference | Non-pregnant | Pregnant | Difference |
|---|---|---|---|---|---|---|---|---|---|
| IV (ZT18–ZT24) | 1 | 100 | 100 | 22.33 [22.24, 22.40] | 22.53 [22.45, 22.63] | **0.20 [0.09, 0.33]** | 20.7 [17.6, 24.0] | 9.34 [7.75, 11.0] | **−11.3 [−15.1, −7.67]** |
| | 2 | 100 | 100 | 22.37 [22.27, 22.46] | 22.50 [22.42, 22.60] | **0.13 [0.01, 0.27]** | 22.5 [18.7, 26.5] | 10.5 [8.56, 12.5] | **−12.0 [−16.5, −7.61]** |
| | 3 | 100 | 100 | 22.27 [22.10, 22.39] | 22.43 [22.20, 22.70] | 0.17 [−0.09, 0.47] | 20.5 [16.5, 24.9] | 7.19 [5.29, 9.26] | **−13.3 [−18.2, −8.71]** |
| V (ZT24–ZT4) | 1 | 97.1 | 99.8 | 1.57 [1.25, 3.73] | 2.91 [2.51, 3.27] | 1.34 [−0.64, 1.83] | 6.59 [4.81, 8.51] | 4.07 [3.04, 5.21] | **−2.52 [−4.78, −0.37]** |
| | 2 | 66.3 | 83.9 | 2.10 [1.19, 3.94] | 3.18 [1.91, 3.93] | 1.09 [−0.93, 2.45] | 4.46 [3.02, 6.14] | 3.01 [2.07, 4.13] | −1.44 [−3.46, 0.43] |
| | 3 | 87.7 | 38.5 | 2.03 [1.29, 3.86] | 3.53 [1.44, 3.97] | 1.51 [−0.48, 2.57] | 5.25 [3.45, 7.26] | 2.34 [1.44, 3.44] | **−2.91 [−0.52, −0.80]** |

Abbreviations: CrI, 95% credible interval; study block = block 1 (days 0.5–6.5, *n* = 31), block 2 (days 6.5–12.5, *n* = 21), block 3 (days 12.5–17.5, *n* = 11); non-pregnant mice, *n* ≥ 6; ZT, Zeitgeber. Intervals of group differences that exclude zero are shown in bold text. Positive values present delayed timing or increased amplitude. Negative values present advanced timing or decreased amplitude.

the pregnant group exhibited a peak in activity (Table 4). Within mice that exhibited a peak between ZT8 and ZT12, the timing and amplitudes in both groups were similar (Table 4). In the second, third, fourth and fifth time periods of interest (ZT12–ZT15, ZT15–ZT18, ZT18–ZT24 and ZT24–ZT4), 95, 47, 100 and 87% of the non-pregnant group and 62, 76, 100 and 38% of the pregnant group, respectively, exhibited activity peaks (Table 3). Across these time periods, the peak amplitude was lower (0.7-, 0.8-, 0.6- and 0.6-fold, respectively) in the pregnant group than in the non-pregnant group, and the timing of peaks was similar between groups (Table 3).

### Effect of pregnancy on wakefulness

Modelled probabilities of wakefulness corresponded to raw data for time spent awake in each of the three blocks of the study (Fig. 5*A–C*). Modelled probabilities of being awake were similar in pregnant and non-pregnant mice throughout the light period and early and late in the dark periods (ZT22.5–ZT13.5) across all days of the study (Fig. 5*D–F*). The probability of being awake between ZT13.5 and ZT22.5 was lower in pre-gnant than non-pregnant mice in days 0.5–6.5 and 6.5–12.5 of study (Fig. 5*D* and *E*). During days 12.5–17.5, the probability of being awake was lower in pregnant than non-pregnant mice between ZT13.5–ZT20 and ZT21.5–ZT22.5 (Fig. 5*F*).

**Days 0.5–6.5.** Within the first and fifth time periods of interest (ZT8–ZT12 and ZT24–ZT4), 68 and 98% of the non-pregnant group and 64 and 98% of the pregnant group, respectively, exhibited peaks in wakefulness that did not differ in timing or amplitude (Table 4). In the second, third and fourth time periods of interest (ZT12–ZT15, ZT15–ZT18 and ZT18–ZT24), >99% of the non-pregnant group and >95% of the pregnant group exhibited peaks in wakefulness (Table 4). The probability of being awake was lower (0.1-, 0.4- and 0.2-fold, respectively) in the pregnant group than in the non-pregnant group for all these time periods during the dark period (Table 4). The timing of peaks in wakefulness between ZT12 and ZT15 and between ZT15 and ZT18 did not differ between groups, whilst the peak in wakefulness in the second half of the dark period (ZT18–ZT24) was 0.11 h later in the pregnant than non-pregnant group (Table 4).

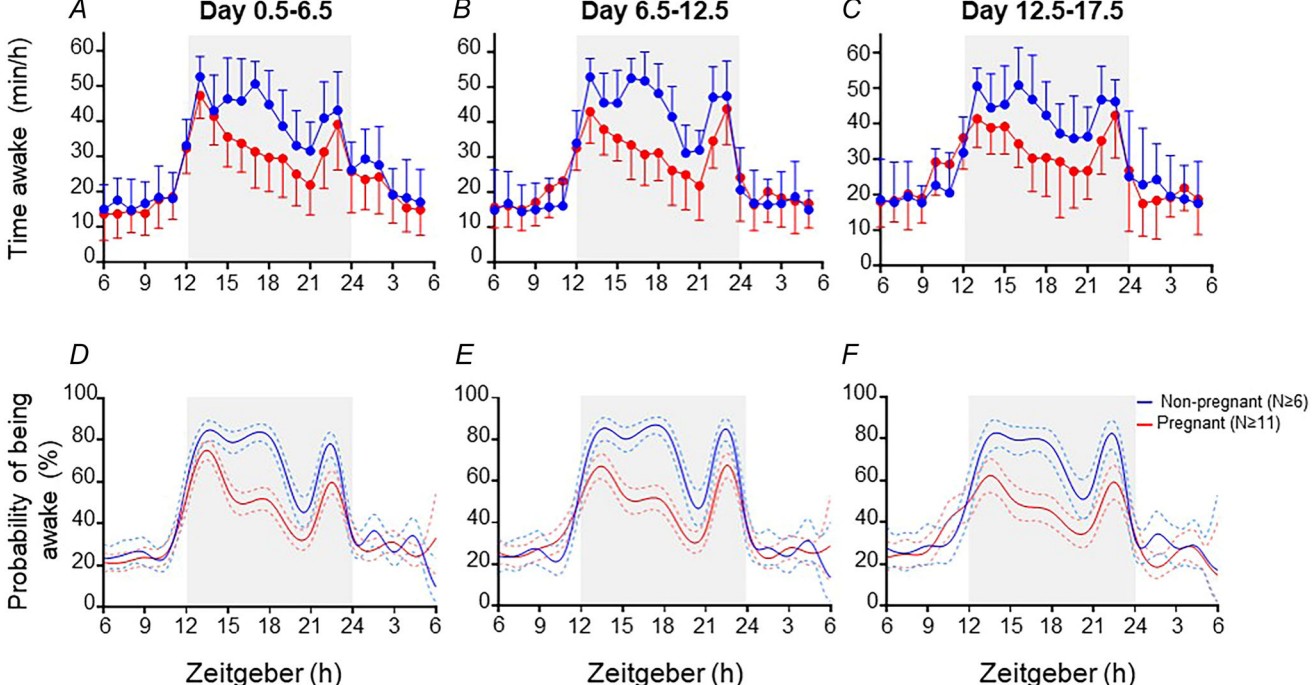

**Figure 5. Sleep–wake behaviour of non-pregnant and pregnant mice**
Time spent awake (*A–C*) is the mean (SD) of raw data for each mouse, averaged across each study block (day 0.5–6.5, 6.5–12.5 and 12.5–17.5), within non-pregnant (blue line, *n* ≥ 6) and pregnant (red line, *n* ≥ 11) groups. The light phase (ZT6–ZT12 and ZT24–ZT6, no shading) and dark phase (ZT12–ZT24, shaded) are shown for day 0.5–6.5 (*A*), day 6.5–12.5 (*B*) and day 12.5–17.5 (*C*). The fitted model of sleep probability (*D–F*) across each of the three blocks of the study indicates fitted means (continuous lines) and 95% credible intervals (dashed lines) for all non-pregnant (blue) and pregnant (red) groups. The light phase (ZT6–ZT12 and ZT24–ZT6, no shading) and dark phase (ZT12–ZT24, shaded) are shown for day 0.5–6.5 (*D*), day 6.5–12.5 (*E*) and day 12.5–17.5 (*F*).

**Table 4. The effect of pregnancy on timing and amplitudes of peaks in probability of being awake**

| Time period of interest | Study block | Iterations per group with detected peak (%) | | Peak characteristics for mice with detected peak | | | | | |
| --- | --- | --- | --- | --- | --- | --- | --- | --- | --- |
| | | | | Peak time (ZT) [95% CrI] | | | Peak amplitude (%) [95% CrI] | | |
| | | Non-pregnant | Pregnant | Non-pregnant | Pregnant | Difference | Non-pregnant | Pregnant | Difference |
| I (ZT8–ZT12) | 1 | 68.8 | 64.0 | 8.67 [8.08, 9.10] | 8.88 [8.10, 9.32] | 0.21 [−0.64, 0.96] | 26.8 [20.6, 33.4] | 23.7 [19.3, 28.5] | −3.1 [−11.1, 4.8] |
| | 2 | 74.9 | 25.6 | 8.70 [8.10, 9.10] | 9.13 [8.24, 9.53] | 0.41 [−0.51, 1.13] | 27.6 [20.0, 35.9] | 28.0 [22.5, 33.8] | 0.39 [−9.8, 8.9] |
| | 3 | 54.4 | 20.4 | 9.05 [8.16, 10.62] | 10.77 [8.0, 11.27] | 1.71 [−0.22, 2.77] | 28.8 [19.7, 39.1] | 43.0 [35.1, 51.4] | **14.2 [1.2, 27.0]** |
| II (ZT12–ZT15) | 1 | 99.1 | 100 | 13.71 [13.49, 14.02] | 13.47 [13.35, 13.56] | −0.24 [−0.56, 0.00] | 84.7 [79.5, 89.2] | 75.0 [70.5, 79.5] | **−9.67 [−16.4, −2.9]** |
| | 2 | 96.8 | 100 | 13.70 [13.42, 14.07] | 13.41 [13.10, 13.59] | −0.29 [−0.74, 0.06] | 85.4 [79.5, 90.5] | 67.2 [61.1, 72.9] | **−18.2 [−26.6, −9.9]** |
| | 3 | 87.1 | 99.9 | 13.97 [13.62, 14.63] | 13.51 [12.86, 13.86] | −0.46 [−1.28, 0.09] | 82.7 [75.6, 88.8] | 62.3 [54.2, 70.3] | **−20.4 [−30.6, −9.5]** |
| III (ZT15–ZT18) | 1 | 99.8 | 95.9 | 17.26 [14.69, 17.57] | 17.46 [17.06, 17.71] | 0.21 [−0.31, 0.85] | 83.8 [79.4, 87.6] | 51.9 [46.5, 57.2] | **−31.9 [−38.5, −25.1]** |
| | 2 | 99.9 | 86.2 | 17.33 [16.80, 17.63] | 17.23 [16.71, 17.59] | −0.09 [−0.73, 0.56] | 87.0 [82.7, 90.7] | 52.1 [46.4, 57.9] | **−35.0 [−42.0, −27.7]** |
| | 3 | 73.9 | 46.7 | 16.87 [15.32, 17.58] | 17.35 [16.67, 17.83] | 0.47 [−0.54, 2.08] | 79.9 [74.1, 85.2] | 46.6 [39.9, 53.2] | **−33.3 [−41.9, −24.3]** |

*(Continued)*

**Table 4. (Continued)**

| Time period of interest | Study block | Iterations per group with detected peak (%) | | Peak characteristics for mice with detected peak | | | | | |
| | | | | Peak time (ZT) [95% Crl] | | | Peak amplitude (%) [95% Crl] | | |
| | | Non-pregnant | Pregnant | Non-pregnant | Pregnant | Difference | Non-pregnant | Pregnant | Difference |
| IV (ZT18–ZT24) | 1 | 100 | 100 | 22.38 [22.31, 22.45] | 22.50 [22.42, 22.57] | **0.11 [0.02, 0.22]** | 78.1 [71.8, 83.7] | 59.8 [54.1, 65.3] | **−18.4 [−26.6, −10.0]** |
| | 2 | 100 | 100 | 22.43 [22.35, 22.50] | 22.54 [22.46, 22.62] | **0.11 [0.01, 0.22]** | 85.1 [79.2, 90.0] | 67.6 [61.6, 73.4] | **−17.5 [−25.4, −9.2]** |
| | 3 | 100 | 100 | 22.33 [22.22, 22.42] | 22.49 [22.35, 22.68] | 0.16 [−0.01, 0.37] | 82.4 [74.8, 88.7] | 59.2 [50.8, 67.2] | **−23.2 [−34.0, −12.1]** |
| V (ZT24–ZT4) | 1 | 98.9 | 98.9 | 1.59 [1.35, 1.99] | 2.72 [1.66, 3.32] | 1.13 [−0.06, 1.82] | 36.5 [29.2, 44.3] | 31.1 [25.9, 36.5] | −5.44 [−15.2, 3.91] |
| | 2 | 72.1 | 82.4 | 1.86 [1.20, 3.92] | 3.17 [2.50, 3.92] | 1.30 [−0.80, 2.45] | 27.0 [19.7, 35.1] | 28.1 [22.9, 33.8] | 1.02 [−9.0, 10.6] |
| | 3 | 97.3 | 59.1 | 1.88 [1.36, 3.76] | 3.51 [2.93, 3.97] | 1.64 [−0.33, 2.46] | 34.0 [22.0, 44.1] | 27.9 [21.6, 34.8] | −6.1 [−17.8, 5.74] |

Abbreviations: Crl, 95% credible interval; study block = block 1 (days 0.5–6.5, *n* = 31), block 2 (days 6.5–12.5, *n* = 21), block 3 (days 12.5–17.5, *n* = 11); non-pregnant mice, *n* ≥ 6; ZT, Zeitgeber. Intervals of group differences that exclude zero are shown in bold text. Positive values present delayed timing or increased amplitude. Negative values present advanced timing or decreased amplitude.

**Days 6.5–12.5.** During the light phase [first (ZT8–ZT12) and fifth time periods of interest (ZT24–ZT4)], 74 and 72% of the non-pregnant group and 25 and 82% of the pregnant group, respectively, exhibited peaks in wakefulness, and these were similar in timing and amplitude between groups (Table 4). During the dark phase [second (ZT12–ZT15), third (ZT15–ZT18) and fourth time periods of interest (ZT18–ZT24)], >96% of the non-pregnant group and 100, 86 and 100% of the pregnant group, respectively, exhibited wakefulness peaks (Table 4). The probability of being awake was lower (0.2-, 0.4- and 0.2-fold, respectively) in the pregnant group than in the non-pregnant group across these time periods (Table 4). The timing of peaks in wakefulness for the second and third time periods of interest did not differ between groups, and the peak in wakefulness in the second half of the dark period (ZT18–ZT24) was 0.11 h later in the pregnant group than in the non-pregnant group (Table 4).

**Days 12.5–17.5.** Within the first time period of interest (ZT8–ZT12), 54% of the non-pregnant group and 20% of the pregnant group exhibited a peak in wakefulness (Table 4). Within mice that exhibited a peak in the probability of wakefulness, the amplitude of wakefulness was 1.5-fold greater in the pregnant group than in the non-pregnant group, but the peaks occurred at similar times (Table 4). Throughout the dark phase, in the second, third and fourth time periods of interest (ZT12–ZT15, ZT15–ZT18 and ZT18–ZT24) 87, 73 and 100% of the non-pregnant group and 99, 46 and 100% of the pregnant group, respectively, exhibited wakefulness peaks (Table 4). Within mice that exhibited a peak in behaviour, the probability of being awake was lower (0.2-, 0.4- and 0.3-fold, respectively) in the pregnant than non-pregnant group for each of these time periods, but the timing of the peaks did not differ between groups (Table 4). Within the fifth time period of interest (ZT24–ZT4) 97% of the non-pregnant group and 59% of the pregnant group exhibited a peak in wakefulness, and there was no difference in timing or amplitude of peaks within mice that exhibited this peak (Table 4).

## Discussion

We identified that mice exhibit diurnal patterns in food and water intake, activity and wakefulness behaviours, regardless of pregnancy status. In this study, the circadian pattern of behaviour during pregnancy, relative to non-pregnant mice, was characterised by: (i) a reduction in physical activity and in time spent awake during the dark phase, beginning during days 0.5–6.5 of pregnancy; (ii) an increase in food intake at the end of the light phase during days 6.5–12.5 and 12.5–17.5 of pregnancy; and (iii) an increase in food and water intake at the start of the dark phase during the final 12.5–17.5 days of pregnancy. Consistent with some (Yaw et al., 2021) but not all (Martin-Fairey et al., 2019) previous reports, we observed delays in circadian behaviours, evidenced by a later onset of dark-phase activity and by later peaks in behaviours within the groups that displayed peaks in time periods of interest, particularly at the end of the dark phase (activity and wakefulness) and end of the light phase (food intake). Thus, both the timing and total amounts of each behaviour are altered during mouse pregnancy.

## Changes in food and water intake behaviour during pregnancy

Rodents used in biomedical research are nocturnal and consume ∼65–80% of their daily food (Christie et al., 2018; Johnson & Johnson, 1990) and 78–90% of their daily water in the active dark phase (Johnson & Johnson, 1990; Oatley, 1971). In the present study, we replicate with our data that food and water intake followed strong circadian patterns aligning with the sleep–wake cycle. Furthermore, our data, together with others, shows conservation of nocturnally dominated patterns of food intake during pregnancy in rodents. We also observed that the timing of water intake in the pregnant and non-pregnant female groups was closely linked to peaks in eating. This is also consistent with reports that in male rats 70% of water intake is associated with food intake (Johnson & Johnson, 1990) and, more specifically, that 57% of total water consumption occurs within 20 min of eating (Oatley, 1971).

Although the nocturnal bias in food intake was preserved, the timing of food intake changed during pregnancy. We have reported previously that the increased food intake during mouse pregnancy (Ladyman et al., 2018; Li et al., 2021; Neubauer & Mletzko, 1990) reflects increased food intake during the light phase, owing to larger meal size, rather than substantial increases in dark-phase food intake (Li et al., 2021). In the present study, we identified that this increase in light-phase food intake during pregnancy was attributable to pregnant mice consuming a greater amount of food shortly before the dark phase. Interestingly, in groups that exhibited a peak in the last 4 h of the light phase (ZT8–ZT12), this peak was delayed in comparison to the peak observed in the non-pregnant group from days 6.5 to 12.5 of pregnancy, the first evidence for delayed feeding behaviours during pregnancy. Furthermore, a similar delay in water intake behaviours was also observed during the second block, day 6.5–12.5, of pregnancy. Although the timing of food and water intake peaks early in the dark period (second time period of interest, ZT12–ZT15), at times when food intake is greater in male non-pregnant mice (Christie

et al., 2018), was unaltered by pregnancy, the timing of maximal food intake differed. Our observation that more of the pregnant group exhibited a peak in food and water earlier in the dark phase, during the second time period of interest (ZT12–ZT15), whereas the non-pregnant group consumed food between the second and third time periods of interest (ZT12–ZT15 and ZT15–ZT18), is consistent with changes in the timing of maximal food intake in rat pregnancy (Neubauer & Mletzko, 1990). Maximal food intake occurs within a shorter feeding window in pregnant than non-pregnant rats (ZT11–ZT15 cf. ZT11–ZT19), in which food intake events occur within two peaks during pregnancy rather than a single peak in non-pregnant rats (Neubauer & Mletzko, 1990).

The mechanisms underlying changes in food and water intake during pregnancy are not clear, although the delayed onset in days 6.5–12.5 and 12.5–17.5 after mating suggests that rising concentrations of pregnancy hormones might be responsible (Clarke et al., 2021). Although plasma oestrogen increases during mid-pregnancy and remains elevated during late pregnancy in mice, we do not consider this hormone a likely candidate mechanism for advancement of food intake behaviour during mouse pregnancy, because oestradiol does not alter the phase timing of food intake in female rats (Palmisano et al., 2017). Maternal circulating growth hormone (GH) concentrations also increase by mid-pregnancy and remain elevated in the pregnant mouse (Gatford et al., 2017). Although the impact of GH on circadian patterns of food intake has not been assessed directly, administration of GH-releasing factor directly into the brain stimulates food intake during the inactive but not the active phase in male rats and hamsters (Feifel & Vaccarino, 1989; Vaccarino et al., 1995). We therefore hypothesise that elevated maternal GH during mouse pregnancy might underlie increased light-phase food intake. However, this remains to be determined.

### Changes in physical activity and sleep patterns in pregnant mice

From early pregnancy, we observed that mice slept significantly more during the dark phase, replacing the time spent active around the cage. This pattern of rapid reduction in movement around the metabolic cages during pregnancy is consistent with the rapid reduction in voluntary wheel-running activity and increased sleep beginning at the start of pregnancy reported in the same strain of mice by Ladyman et al. (2018). The timing of this rapid decrease in activity and increase in sleep even before implantation, at ∼4 days after mating (Yoshinaga, 2013), implies that the drivers for reduced activity are maternal in origin and do not originate from the fetus or placenta. It has been hypothesised that the reduction in activity in early pregnancy is driven by prolactin

(Ladyman et al., 2018, 2021), because prolactin is one of the first maternal hormones to increase after mating (Phillipps et al., 2020). Although in other studies maternal activity did not decrease until mid-pregnancy in mice (Martin-Fairey et al., 2019; Yaw et al., 2021), this might reflect different methodologies, including activity analysis (Yaw et al., 2021, Ladyman et al.,2018 and present study: 5.0 min bins; Martin-Fairey et al., 2019, 6 min bins) and cage systems (Yaw et al., 2021, clocklab; Martin-Fairey et al., 2019, circadian cabinets; Ladyman et al.,2018 and present study, Promethion cages). Reduced activity late in mouse pregnancy is likely to reflect the impact of both hormonal changes and body weight, because maternal body weight increases rapidly in the second half of pregnancy. At the end of the present study, pregnant mice at GD17.5 were 55% heavier than their age-matched non-pregnant controls (Li et al 2021).

In contrast to the consistent reports of decreased activity during pregnancy in mice, the reported changes in timing of activity are inconsistent. In the present study, pregnant mice took longer to reach their activity threshold as pregnancy progressed, whereby we observed a 1.86 (1.99) h delay in the time to reach activity onset in the dark phase on day 1 and a 6.07 (5.17) h delay on day 17, in pregnant compared with non-pregnant mice. Furthermore, the activity and wakefulness peak that occurred late in the dark period was also delayed on days 0.5–6.5 and 6.5–12.5 in pregnancy, although to a lesser extent (11–20 min), and did not persist in late pregnancy. This is the first detailed report of circadian patterns of activity and time spent awake during mouse pregnancy. The onset of activity in running wheels occurs early in the dark period in male and non-pregnant female mice (reviewed by Bains et al., 2018), and is used as a single daily measure of activity timing. Data on the timing of running wheel activity in pregnancy is inconsistent. Yaw et al. (2021) reported delayed running wheel onset in mid-pregnant (GD8–GD13) compared with non-, early- and late-pregnant mice. Conversely, Martin-Fairey et al. (2019) reported earlier (≤4 h) running wheel activity between GD3 and GD10, relative to non-pregnant controls, whereas in later pregnancy the timing of activity returned to that of non-pregnant mice. Effects of pregnancy on different types of behaviour might differ with experimental factors, such as the measure of activity. Given that total cage movement reflects spontaneous activity, whereas mice find running wheel exercise rewarding (Novak et al., 2012), changes in wheel-running activity might reflect altered reward motivation. Furthermore, increased body size in late pregnancy might restrict the ability of mice to access the wheel in late pregnancy and confound measures of activity using this approach. Additionally, a strength of the present modelling approach is capturing the timing and amplitude of multiple peaks in each behaviour throughout the circadian cycle and assessing whether this differs

between groups. This approach is a novel aspect of the statistical design and provides greater insight into the effects of pregnancy on behavioural patterns rather than single onset events.

The mechanisms underlying the delay in activity and wakefulness during pregnancy are unknown, although there is evidence for impacts of both progesterone and oestrogen. Progesterone levels increase from GD4 in the mouse and remain relatively high for the duration of pregnancy (Piekorz et al., 2005). In cycling female rats, progesterone implants delayed the onset of wheel-running activity at the start of the dark phase by 22 min in comparison to cholesterol-implanted controls (Albers et al., 1981). Albers et al. (1981) hypothesised that progesterone antagonises oestrogen, because increases in oestrogen in female rats during pro-oestrus occurred concurrent with advanced activity onset and increased wheel-running activity (Albers et al., 1981). Likewise, oestrogen implants in hamsters advanced the timing of activity onset and consolidated activity bouts to earlier in the active phase (Morin et al., 1977). It is therefore likely that activity and sleep are altered by a combination of rising progesterone and prolactin in early pregnancy and that progesterone antagonizes oestrogen in later pregnancy to delay activity onset, but these hypotheses are yet to be tested.

## Conclusion

This study confirms and extends previous observations that normal circadian rhythms of behaviour are altered during pregnancy, and the differences in the timing and amplitude of each behaviour are likely to reflect the role of different pregnancy hormones. Increased food intake at the start of the light phase and end of the dark phase during pregnancy reflects increased amplitude of eating behaviour, without longer duration. Marked decreases in activity and the probability of being awake also contribute to positive energy balance in pregnancy, with delays to all measured behaviours being evident from mid-pregnancy onwards. Further research is required to determine whether pregnancy complications, observed for example in maternal obesity, result from disruption in these adaptations in circadian behaviour.

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

## Additional information

### Data availability statement

Data have been made available on figshare (https://doi.org/10.25909/c.7111231.v1)

### Competing interests

The authors declare no conflicts of interest.

### Author contributions

G.S.C., K.L.G. and A.J.P. designed the study; G.S.C. and A.D.V. analysed the data; G.S.C. wrote the manuscript; all authors were involved in data interpretation and editing the manuscript, approved the final version and agree to be accountable for all aspects of the work in ensuring that questions related to the accuracy or integrity of any part of the work are appropriately investigated and resolved. All persons designated as authors qualify for authorship, and all those who qualify for authorship are listed.

### Funding

G. S. Clarke held the Robinson Honours scholarship of the Robinson Research Institute, University of Adelaide, and is now supported by an Australian Government Research Training Program (RTP) Stipend Scholarship.

### Acknowledgements

Open access publishing facilitated by The University of Adelaide, as part of the Wiley - The University of Adelaide agreement via the Council of Australian University Librarians.

### Keywords

circadian, food intake, physical activity, pregnancy

### Supporting information

Additional supporting information can be found online in the Supporting Information section at the end of the HTML view of the article. Supporting information files available:

**Peer Review History**

