## [Peer Review History · The Journal of Physiology]

Circadian patterns of behaviour change during pregnancy in mice

Georgia S Clarke, Andrew Vincent, Sharon R Ladyman, Kathryn L Gatford, and Amanda J Page
DOI: 10.1113/JP285553

Corresponding author(s): Georgia Clarke (georgia.clarke@adelaide.edu.au)

The following individual(s) involved in review of this submission have agreed to reveal their identity: Erik D Herzog (Referee #1); Sean Williams (Referee #3)

Review Timeline:

Submission Date:	24-Aug-2023
Editorial Decision:	05-Oct-2023
Revision Received:	17-Nov-2023
Editorial Decision:	30-Nov-2023
Revision Received:	31-Jan-2024
Accepted:	22-Feb-2024

Senior Editor: Paul Greenhaff

Reviewing Editor: Josiane Broussard

Transaction Report:

Dear Dr Clarke,

Re: JP-RP-2023-285553 "Circadian patterns of behaviour change during pregnancy in mice" by Georgia S Clarke, Andrew Vincent, Sharon R Ladyman, Kathryn L Gaford, and Amanda J Page

Thank you for submitting your manuscript to The Journal of Physiology. It has been assessed by a Reviewing Editor and by 3 expert referees and we are pleased to tell you that it is potentially acceptable for publication following satisfactory major revision.

REVISION CHECKLIST:

We look forward to receiving your revised submission.

Yours sincerely,

Paul Greenhaff
Senior Editor
The Journal of Physiology

REQUIRED ITEMS

- Author photo and profile. First (or joint first) authors are asked to provide a short biography (no more than 100 words for one author or 150 words in total for joint first authors) and a portrait photograph. These should be uploaded and clearly labelled with the revised version of the manuscript. See Information for Authors for further details.
- You must start the Methods section with a paragraph headed Ethical Approval. A detailed explanation of journal policy and regulations on animal experimentation is given in Principles and standards for reporting animal experiments in The Journal of Physiology and Experimental Physiology by David Grundy *J Physiol*, 593: 2547-2549. doi:10.1113/JP270818.). A checklist outlining these requirements and detailing the information that must be provided in the paper can be found at: <https://physoc.onlinelibrary.wiley.com/hub/animal-experiments>. Authors should confirm in their Methods section that their experiments were carried out according to the guidelines laid down by their institution's animal welfare committee, and conform to the principles and regulations as described in the Editorial by Grundy (2015). The Methods section must contain details of the anaesthetic regime: anaesthetic used, dose and route of administration and method of killing the experimental animals.
- Your manuscript must include a complete Additional Information section.
- The Journal of Physiology funds authors of provisionally accepted papers to use the premium BioRender site to create high resolution schematic figures. Follow this link and enter your details and the manuscript number to create and download figures. Upload these as the figure files for your revised submission. If you choose not to take up this offer we require figures to be of similar quality and resolution. If you are opting out of this service to authors, state this in the Comments section on the Detailed Information page of the submission form. The link provided should only be used for the purposes of this submission. Authors will be charged for figures created on this premium BioRender account if they are not related to this manuscript submission.
- Please upload separate high-quality figure files via the submission form.
- Please ensure that any tables are in Word format and are, wherever possible, embedded in the article file itself.
- Please ensure that the Article File you upload is a Word file.
- Papers must comply with the Statistics Policy: https://jp.msubmit.net/cgi-bin/main.plex?form_type=display_requirements#statistics.

In summary:

- If $n \leq 30$, all data points must be plotted in the figure in a way that reveals their range and distribution. A bar graph with data points overlaid, a box and whisker plot or a violin plot (preferably with data points included) are acceptable formats.
- If $n > 30$, then the entire raw dataset must be made available either as supporting information, or hosted on a not-for-profit repository e.g. FigShare, with access details provided in the manuscript.
- 'n' clearly defined (e.g. x cells from y slices in z animals) in the Methods. Authors should be mindful of pseudoreplication.
- All relevant 'n' values must be clearly stated in the main text, figures and tables.

- The most appropriate summary statistic (e.g. mean or median and standard deviation) must be used. Standard Error of the Mean (SEM) alone is not permitted, unless justified and presented alongside confidence intervals.

- Exact p values must be stated. Authors must not use 'greater than' or 'less than'. Exact p values must be stated to three significant figures even when 'no statistical significance' is claimed.

- Please include an Abstract Figure file, as well as the figure legend text within the main article file. The Abstract Figure is a piece of artwork designed to give readers an immediate understanding of the research and should summarise the main conclusions. If possible, the image should be easily 'readable' from left to right or top to bottom. It should show the physiological relevance of the manuscript so readers can assess the importance and content of its findings. Abstract Figures should not merely recapitulate other figures in the manuscript. Please try to keep the diagram as simple as possible and without superfluous information that may distract from the main conclusion(s). Abstract Figures must be provided by authors no later than the revised manuscript stage and should be uploaded as a separate file during online submission labelled as File Type 'Abstract Figure'. Please ensure that you include the figure legend in the main article file. All Abstract Figures should be created using BioRender. Authors should use The Journal's premium BioRender account to export high-resolution images. Details on how to use and access the premium account are included as part of this email.

EDITOR COMMENTS

Reviewing Editor:

Authors investigated the timely (no pun intended) topic of daily rhythms in pregnant mice. Reviewers found the findings intriguing and to be of interest to a variety of investigators. The manuscript is well written and the experiments were executed. Reviewers note some suggestions that will improve the manuscript to help ensure a bigger impact.

Senior Editor:

This manuscript has been considered by a Reviewing Editor, Statistics Editor and two specialist reviewers who were enthusiastic about the work. The study is however descriptive in nature and it is unclear what the "modelling" is bringing that is not available in the mean \pm SD data. Moreover, there is a query from the Statistics Editor about the power of the study and the statistical methods employed. The abstract states animals were "early- (N = 10), mid- (N = 10) or late-pregnancy (N = 11), or as age-matched, non-pregnant controls (N = 12) please therefore include the 'n' values (animal numbers) in Figs 1 - 4 at each week in each group. Secondly, what findings are revealed when the mean and SD data in Figs 1 - 4 are compared using conventional statistical approaches such as two-way ANOVA (time and treatment). At the moment I do not understand the manner in which the data are presented or interpreted by the authors, which needs serious consideration in any revision.

REFEREE COMMENTS

Referee #1:

The authors tested the hypothesis that daily rhythms change in pregnant mice. This is timely since two recent publications examining locomotor activity found differing effects of early pregnancy on the onset and level of locomotor activity. The authors take advantage of a data set collected during a prior 2021 electrophysiology study and report modest, but significant, changes in daily patterns of food and water intake, locomotor activity, and time spent awake. The changes are intriguing and will be of interest to a variety of researchers and, perhaps, clinicians. The manuscript is well written and the experiments were executed. The authors use appropriate statistics. The manuscript will be improved when the authors address these concerns.

1. The authors claim to see changes in daily patterns that differ from prior reports (e.g., Martin-Fairey et al., 2019 and Yaw et al., 2021) but have analyzed their data in a unique way. In addition to providing hourly averages from each week of pregnancy, the paper would benefit from presenting the hourly average across mice throughout pregnancy (e.g., as in Martin-Fairey et al. 2019). As presented, we lack a justification for treating the first, second and third weeks of pregnancy as categories during pregnancy for analysis of daily rhythms.

2. Can the authors explain what is learned by "modeling" their data (e.g. Fig 1D-F; 2D-F; etc.)? The models appear to be a way of representing the same data without furthering insight into the underlying mechanisms or consequences of changes presented with the data averaged across mice.

Referee #2:

This study aimed to assess circadian rhythm of food and water intake, activity and wakefulness during pregnancy compared

to non-pregnant control mice. Some findings are new and some findings are new additions to the literature. The experiments are well done. Results provide new insight into understanding how pregnancy might influence behaviors.

My only comment is that in order to indicate the results are from healthy pregnancy and the differences are normal adaptations to pregnancy, the authors need to present evidence that these are indeed healthy pregnancy.

Referee #3:

The statistical analyses performed are complex, perhaps overly so given the descriptive and relatively low-power nature of the study. Whilst I'm confident the analyses have been performed in an accurate and robust manner, it would be helpful for the authors to justify their statistical choices and outline the benefits of this multi-level Bayesian approach over and above the more traditional two-way ANOVA (time X group) approach (both in their responses here and in the manuscript). I presume this relates to their ability to account for the unbalanced structure of the dataset (different N's at each time point) and account for individual differences in responses, but it would be helpful for the authors to outline this along with any other advantages. Making the code and data for this analysis available as a Supplementary file would also aid the reviewers and eventual readers in understanding the approach taken.

The decision to square-root transform the data should be justified in the manuscript - why was this necessary? The authors also need to state in the Methods how decisions about 'significant' effects were made i.e., was it purely when intervals for group differences excluded 0?

As per the Journal's statistics policy (https://jp.msubmit.net/cgi-bin/main.plex?form_type=display_requirements#statistics) the relevant 'N' should be stated in the figures, tables, and their legends.

In Table 1, the peak time difference for Weeks 2 and 3 in time period 1 are bolded: 1.62 [-2.22,0.70], 1.60 [-2.47, 0.07] but these do not exclude zero. This finding is stated in the Abstract and elsewhere and presented as 'significant', so please check and clarify whether this is accurate.

END OF COMMENTS

Confidential Review

24-Aug-2023

We thank the editors and reviewers for their time to consider our paper and for providing these valuable comments. We have addressed the reviewer suggestions to improve the manuscript as detailed below. Line and page numbers are for the “tracked changes” version of the revised paper.

Comment from reviewing editor:

Authors investigated the timely (no pun intended) topic of daily rhythms in pregnant mice. Reviewers found the findings intriguing and to be of interest to a variety of investigators. The manuscript is well written and the experiments were executed. Reviewers note some suggestions that will improve the manuscript to help ensure a bigger impact.

Response to referee 1:

The authors tested the hypothesis that daily rhythms change in pregnant mice. This is timely since two recent publications examining locomotor activity found differing effects of early pregnancy on the onset and level of locomotor activity. The authors take advantage of a data set collected during a prior 2021 electrophysiology study and report modest, but significant, changes in daily patterns of food and water intake, locomotor activity, and time spent awake. The changes are intriguing and will be of interest to a variety of researchers and, perhaps, clinicians. The manuscript is well written and the experiments were executed. The authors use appropriate statistics. The manuscript will be improved when the authors address these concerns.

1. a) The authors claim to see changes in daily patterns that differ from prior reports (e.g., Martin-Fairey et al., 2019 and Yaw et al., 2021) but have analyzed their data in a unique way. In addition to providing hourly averages from each week of pregnancy, the paper would benefit from presenting the hourly average across mice throughout pregnancy (e.g., as in Martin-Fairey et al. 2019). b) As presented, we lack a justification for treating the first, second and third weeks of pregnancy as categories during pregnancy for analysis of daily rhythms.

a) We suspect that the differences in changes in daily patterns in the present study, compared to prior reports is likely due to the type of activity data recorded. Within both Martin-Fairey et al., 2019 and Yaw et al., 2021 studies, the daily onset of activity was defined as the first time when activity was counted for at least 1 hour after at least 4 hours of inactivity and activity was measured using a running wheel. Within our cohort, we did not equip the metabolic cages with running wheels, and instead analysed spontaneous movement of mice around the cage. As discussed [line 495-501, page 17-18] we believe this avoids some limitations of using running wheels including potential changes in reward from exercise and the potential for increasing maternal size to restrict running wheel access by late pregnancy. We could not apply the same modelling as in the previous papers because there was never a 4 hour period of inactivity in spontaneous movement. Instead we have modelled the observed behaviour in a manner that allows for identification of differences between pregnant and non-pregnant groups. .

b) Why we treated the first, second and third weeks of pregnancy as categories during pregnancy for analysis of daily rhythms: Initially we explored models of behaviour change that varied continuously over time (days-since-pregnancy, DSP), spline (time, hr) x group x linear (DSP). However, it became apparent that non-linear modelling for the third component (DSP) of the three-way interaction would be required. The three-way interaction: spline (time, hr) x group x spline (DSP) had too many parameters, and models using polynomials for the third term resulted in poor estimation at the extremes (day 0 and day 17). Hence, we present a simpler model with DSP categorised into three equal periods. These time points correspond to key developmental stages of blastocyst formation and placental development (Panja & Paria, 2021).

We have added this information into the methods:

“Initially we explored models of behaviour change that varied continuously over time (days-since-pregnancy (DSP)), spline (time, hr) x group x linear (days-since-pregnancy). However, it became apparent that non-linear modelling for the third component (DSP) would be required. The three-way interaction: spline (time, hr) x group x spline (DSP) had too many parameters, and models using polynomials for the third term resulted in poor estimation at the extremes (day 0 and day 17). Hence, we present a simpler model with DSP discretised into three categories, namely weeks 1, 2 and 3. These time points represent stages of developmental progression including implantation of the blastocyst and placental development such that, day 5-8: blastocyst implantation site grows, day 10-11: definite placenta structure present and day 15-17: placenta at maximum size (Panja & Paria, 2021)”

[Line 202-2011, page 7-8]

2. Can the authors explain what is learned by "modeling" their data (e.g. Fig 1D-F; 2D-F; etc.)? The models appear to be a way of representing the same data without furthering insight into the underlying mechanisms or consequences of changes presented with the data averaged across mice.

Thank you for this comment. The figures in panels A-C are raw data (mean \pm SD). Our modelling aimed to capture the variation in behaviour observed in the raw data to allow for assessment of when this differs between groups and across days-since-pregnancy. Our modelling is not a mechanistic model, but rather an exploratory model allowing for identification of differences in behaviour between groups and over time.

We have added this into the methods:

“Therefore, we explored models of behaviour change that allowed activity to vary continuously over time using splines and aimed to capture the variation in behaviour observed in the raw data. This model is not a mechanistic model, but rather an exploratory model and allowed us to identify differences in behaviour between groups and over time (days-since-pregnant (DSP)).”

[line 198-202, page 7]

Response to referee 2:

This study aimed to assess circadian rhythm of food and water intake, activity and wakefulness during pregnancy compared to non-pregnant control mice. Some findings are new and some findings are new additions to the literature. The experiments are well done. Results provide new insight into understanding how pregnancy might influence behaviors.

1. My only comment is that in order to indicate the results are from healthy pregnancy and the differences are normal adaptations to pregnancy, the authors need to present evidence that these are indeed healthy pregnancy.

Thank you for this comment. We have provided information on the mouse phenotype (e.g. increases in maternal body weight and litter size) at each pregnancy stage, all of which are useful indicators of a healthy pregnancy in mice [line 242-243, page 9]. Furthermore, the behavioural phenotype of these mice was consistent with another study, utilising Promethion metabolic cages and C57BL/6 mice, also consistent with a healthy pregnancy. This includes increased food intake during mid-pregnancy, due to meal size and duration rather than meal number (our cohort previously reported in (Li *et al.*, 2021) publication compared to (Ladyman *et al.*, 2018)), and physical activity dramatically reduced after mating (current paper and (Ladyman *et al.*, 2018)). We have added a sentence into the methods and results to clarify this for readers.

Methods:

*“The mice were monitored daily and displayed a behavioural phenotype consistent with reports in other healthy pregnancy studies. This includes a significant increase in maternal body weight by day 7, increases in food intake during mid-pregnancy, primarily due to meal size and duration rather than meal number (Ladyman *et al.*, 2018; Li *et al.*, 2021), and a dramatic reduction in physical activity after mating (Ladyman *et al.*, 2018). Fetal number was counted in all pregnancies at termination to ensure fetal number was within the expected range.”*

[Line 149-155, page 6]

Results:

“All mice had between 7 and 11 fetuses, which was within our expectation of what would be seen in a healthy pregnancy, with the exception of one mouse having 4 fetuses.”

[Line 245-246, page 10]

Overall, there was no indication that mice were unhealthy at any stage in the experiment. Our ethical guidelines which we work under required daily monitoring during these experiments, with any sign of a health issue to be reported. None of the mice included in the current study had any health issue flagged during the experiment, including during pregnancy. All authors (except the statistician A.D.V) have +3 years' experience working with pregnant mice and all the data and visual checks of the animals indicated that they were healthy.

Response to referee 3:

1. a) The statistical analyses performed are complex, perhaps overly so given the descriptive and relatively low-power nature of the study. Whilst I'm confident the analyses have been performed in an accurate and robust manner, it would be helpful for the authors to justify their statistical choices and outline the benefits of this multi-level Bayesian approach over and above the more traditional two-way ANOVA (time X group) approach (both in their responses here and in the manuscript). I presume this relates to their ability to account for the unbalanced structure of the dataset (different N's at each time point) and account for individual differences in responses, but it would be helpful for the authors to outline this along with any other advantages. b) Making the code and data for this analysis available as a Supplementary file would also aid the reviewers and eventual readers in understanding the approach taken.

a) Our interest was in the differences in both timing and magnitude of peak activity, an analysis which we believe cannot be performed using the traditional two-way ANOVA (time x group) since the timing of peak activity may differ between groups. This can be addressed by considering time (hr) as a continuous repeated measure, and using splines to allow for non-linear variation in activity within each day with repeated measures per mouse accounted for using a multilevel mixed effects structure. To compare differences between groups (pregnant vs non-pregnant) over the three week period required a three way interaction. We note that this analysis could have been performed with frequentist methodologies, where the differences in the timing of peak activity could have been performed via bootstrapped multilevel mixed effects models. However, model convergence is often a problem for inference via likelihood (frequentist), hence our choice to use the Bayesian Monte Carlo methodology.

We have added this information into the methods:

To compare differences in peak time and location between treatment groups (pregnancy vs non-pregnancy) requires a complex model that allows rapid variation in behaviour and peak location. The traditional two-way ANOVA (time x group) analysis does not allow for assessment of the timing of peak behaviour. Therefore, we explored models of behaviour change that allowed activity to vary continuously over time using splines and aimed to capture the variation in behaviour observed in the raw data."

[line 195-200, page 7]

"We note that the differences in the timing of peak activity could have been analysed using bootstrapped multilevel mixed effects models, however model convergence is often a problem for inference via likelihood, hence our choice was to use the Bayesian Monte Carlo methodology."

[line 225-228, page 8]

b) Thank you for this comment, we have now provided the excel raw data files and R codes on Figshare. If this manuscript is accepted for publication we will make the figshare project published and provide a single reference for readers.

“The raw data and codes for this model have been made available on figshare (link to be added).”

[line 237-238, page 9]

Water intake raw file: <https://figshare.com/s/7605f12eb4df30be0d35>

Food intake raw file: <https://figshare.com/s/332efe0b4011f0b9b3be>

Activity raw file: <https://figshare.com/s/173238a5d7d96d689ed8>

Sleep raw file: <https://figshare.com/s/b15ea3e76ea8e9ed2f31>

Set up +import file: <https://figshare.com/s/a161038a6e260ebe1a46>

Activity analysis file: <https://figshare.com/s/a625e3d3dfc8f9671b82>

Activity code: <https://figshare.com/s/0fc2cae5f47beebc4583>

Sleep analysis file: <https://figshare.com/s/d33a781acfbeda4f687b>

Sleep code: <https://figshare.com/s/7f679570159a148f822b>

Food intake analysis file: <https://figshare.com/s/3e17ed5b88d3012fba8c>

Food intake code: <https://figshare.com/s/4c10a2a6be8c5b89979a>

Water intake analysis file: <https://figshare.com/s/cdde3e88a738437fecae>

Water intake code: <https://figshare.com/s/1541cdd34e5ed395c190>

2. The decision to square-root transform the data should be justified in the manuscript - why was this necessary? The authors also need to state in the Methods how decisions about 'significant' effects were made i.e., was it purely when intervals for group differences excluded 0?

a) All models have assumptions regarding the assumed error distribution. Analyses on the original scale produced residual distributions that were clearly not normally distributed. A square root transformation resolved this problem.

“Due to data distributions we modelled the fraction of time awake/asleep (range 0-1) with a beta distribution. For the other three outcomes (range ≥ 0) analyses on the original scale produced residual distributions that were clearly not normally distributed, a square root transformation was used to resolve this problem.”

[Line 187-190, page 7]

b) We have addressed part b of the comment in the methods:

“Peak time and location were considered statistically significant if the credible interval did not cross over 0.”

[Line 236-237, page 8-9]

3. As per the Journal's statistics policy (https://jp.msubmit.net/cgi-bin/main.plex?form_type=display_requirements#statistics) the relevant 'N' should be stated in the figures, tables, and their legends.

Thank you for this comment, we have amended this.

Please refer to pages 22-23 and 28-31

4. In Table 1, the peak time difference for Weeks 2 and 3 in time period 1 are bolded: 1.62 [-2.22,0.70], 1.60 [-2.47, 0.07] but these do not exclude zero. This finding is stated in the Abstract and elsewhere and presented as 'significant', so please check and clarify whether this is accurate.

Thank you for picking this up, this was a mistake on our behalf. The results were significant and the data was inputted into the table in the incorrect order. It has been amended on page 28.

Gatford KL, Muhlhausler BS, Huang L, Sim PS, Roberts CT, Velhuis JD & Chen C. (2017). Rising maternal circulating GH during murine pregnancy suggests placental regulation. *Endocr Connect* **6**, 260-266.

Ladyman SR, Carter KM & Grattan DR. (2018). Energy homeostasis and running wheel activity during pregnancy in the mouse. *Physiol Behav* **194**, 83-94.

Li H, Clarke GS, Christie S, Ladyman SR, Kentish SJ, Young RL, Gatford KL & Page AJ. (2021). Pregnancy-related plasticity of gastric vagal afferent signals in mice. *Am J Physiol Gastrointest Liver Physiol* **320**, 183-192.

Dear Ms Clarke,

Re: JP-RP-2023-285553R1 "Circadian patterns of behaviour change during pregnancy in mice" by Georgia S Clarke, Andrew Vincent, Sharon R Ladyman, Kathryn L Gatford, and Amanda J Page

Thank you for submitting your manuscript to The Journal of Physiology. It has been carefully assessed by a Reviewing Editor and 2 Referees and relevant feedback from their reports is copied below for your information. We regret to inform you that your manuscript is not considered acceptable for publication in The Journal of Physiology.

In view of reviewer comments, we feel that - even if the specific concerns described below were addressed - the overall priority of this manuscript would not be sufficient for publication in its current form. Although we are unable to offer any further consideration of your manuscript at this stage, we hope that the points raised in the reports will be helpful. You are, of course, welcome to resubmit your manuscript again at a later date if you feel that the concerns raised can be adequately addressed.

We kindly request that you inform your co-authors of this decision as soon as possible.

We know that our decision will be disappointing. We recognize the effort and dedication put into your work, and we deeply appreciate your interest in The Journal of Physiology. Currently, we only accept about 25% of submitted manuscripts, and as a consequence many studies containing good quality research must be declined.

We are sorry that we are unable to communicate a more favourable outcome, but hope that our decision on this occasion will not prevent you from submitting future work to The Journal of Physiology.

Yours sincerely,

Paul Greenhaff
Senior Editor
The Journal of Physiology

EDITOR COMMENTS

Reviewing Editor:

Authors were given the chance to respond to reviewer comments from 2 field experts and a statistical consultation. Reviewers noted that authors have collected exciting data, but major concerns remain regarding the impact and clarity of the findings, as well as the interpretation. Therefore, the manuscript has not moved into a category of high impact that would allow for acceptance/publication.

Senior Editor:

Thank you for the revised version of the manuscript submission to The Journal of Physiology which has been considered by the same specialist reviewers and reviewing editor that considered the original submission. Based on the expert reviewer feedback received the Reviewing Editor is of the opinion that the priority for publication has not increased sufficiently to warrant acceptance for publication in The Journal of Physiology because major concerns remain around the impact and clarity of the findings and the interpretation of data.

REFEREE COMMENTS

Referee #1:

The authors have collected exciting data, but opted to rebut important suggestions that were consistent among the 3 Reviewers. The paper remains intriguing, but the conclusions muddled by weak analysis and confusing text. The major concerns that remain:

1. The authors present each behavior (e.g., water intake) as a daily average profile by averaging data from each dam over a week. This requires them to present averages by week 1, 2, and 3 rather than average daily profiles on day 1, 2, 3, etc. of pregnancy as has been done in prior publications. The resulting comparisons between non-pregnant and pregnant mice do not impress the reader that much has changed.

2. The authors retain models of their data which appear to be spline-smoother versions of the average data. It remains

unclear what we learn from this effort.

Referee #2:

The authors have addressed my previous concern.

1st Confidential Review

17-Nov-2023

Dr Georgia Clarke
School of Biomedicine
University of Adelaide
Vagal Afferent Research Group
Level 7, SAHMRI
North Terrace
Adelaide
E-mail: georgia.clarke@adelaide.edu.au

31 January 2024

Dear Editor-in-Chief, Kim Barnett and Guest Editor, Josianne Broussard,

We thank the editors for accepting our formal appeal against the revision letter and allowing us to further revise the manuscript, “Circadian patterns of behaviour change during pregnancy in mice.”

As requested by the editor and reviewer one, we have analysed the daily changes in activity onset timing and added this to the paper. We have also provided justification for our original analysis of data in three blocks across pregnancy, and further explained the information that this approach provides in addition to the previously reports of once-daily behavioural onset timing during pregnancy by Martin-Fairey *et al.* 2019.

We have addressed each editorial and referee comment below. Page numbers indicating the location of changes are for the “tracked changes” version of the manuscript.

We have also updated the data and coding files for analysis of each outcome (as requested for the last revision of the paper), and placed these on FigShare – these will be published once the manuscript is published.

Water intake raw file: <https://figshare.com/s/7605f12eb4df30be0d35>

Food intake raw file: <https://figshare.com/s/332efe0b4011f0b9b3be>

Activity raw file: <https://figshare.com/s/173238a5d7d96d689ed8>

Sleep raw file: <https://figshare.com/s/b15ea3e76ea8e9ed2f31>

Set up +import file: <https://figshare.com/s/a161038a6e260ebe1a46>

Activity analysis file: <https://figshare.com/s/a625e3d3dfc8f9671b82>

Activity code: <https://figshare.com/s/0fc2cae5f47beebe4583>

Sleep analysis file: <https://figshare.com/s/d33a781acfbeda4f687b>

Sleep R set up: <https://figshare.com/s/4b74535794099ae10532>

Sleep code: <https://figshare.com/s/7f679570159a148f822b>

Food intake analysis file: <https://figshare.com/s/3e17ed5b88d3012fba8c>

Food intake code: <https://figshare.com/s/4c10a2a6be8c5b89979a>

Water intake analysis file: <https://figshare.com/s/cdde3e88a738437fecae>

Water intake code: <https://figshare.com/s/1541cdd34e5ed395c190>

We hope that this revised version and our responses below have addressed the remaining queries to your satisfaction, and that the manuscript is now suitable for publication in the *Journal of Physiology*.

Yours faithfully,

Dr Georgia Clarke

.....

Comment from reviewing editor:

The revised manuscript was considered to be unsuitable for publication because whilst you responded to the comments of Reviewer 2, you unfortunately did not address the concerns to the satisfaction of Reviewer 1 or the Reviewing Editor. This was clearly stated in the response to the authors from the Senior Editor. Essentially you chose to rebut the major concern that Reviewer 1 raised:

1a The authors claim to see changes in daily patterns that differ from prior reports (e.g., Martin-Fairey et al., 2019 and Yaw et al., 2021) but have analyzed their data in a unique way. In addition to providing hourly averages from each week of pregnancy, the paper would benefit from presenting the hourly average across mice throughout pregnancy (e.g., as in Martin-Fairey et al. 2019).

Thank you for this comment. We have added analysis of daily activity onset timing to the revised manuscript, comparing non-pregnant and pregnant mice on each study day, to provide the requested information. In the present study, we analysed spontaneous movement within the cage as our measure of activity, rather than running wheel activity as reported in Martin-Fairey et al., 2019 and Yaw et al., 2021. In the current study mice were not housed with running wheels. We deliberately did not use running wheels to assess activity during pregnancy to avoid potential changes in running wheel activity that might occur late in pregnancy, when mice may find access to the running wheel difficult, and also to avoid the potential effect of pregnancy on brain reward circuitry. We therefore defined activity onset using a different approach to that reported in these previous studies in mice housed with running wheels, where activity onset was defined as the first time when activity (running wheel) was counted for at least 1 hour after at least 4 hours of inactivity.

In order to determine the timing of activity onset based on spontaneous cage activity in our mice, we firstly defined a threshold activity for each individual mouse as the average of dark-phase activity during the acclimatisation period. This approach corrected for inter-individual variation in spontaneous activity prior to mating or pair-housing. The onset of dark-phase activity for each study day was then determined as the first time at which activity exceeded

the threshold activity for the next three consecutive 5 minute blocks. We have added text and a new figure describing our methodology and the results for activity onset time across the study to the paper, as expanded below.

Consistent with these previous murine studies (Martin-Fairey *et al.*, 2019 and Yaw *et al.*, 2021), we used activity as our measure of daily behaviour, and did not analyse sleep onset in mice. Unlike humans, mice are nocturnal and have a polyphasic sleep pattern (Hiyoshi *et al.*, 2014), where they sleep in short minute bursts rather than long hour blocks of complete inactivity. For example, C57Bl/6 mice can have 166 counts of wakefulness episodes, lasting for around 1.5 minutes during the light-phase (Hiyoshi *et al.*, 2014). We therefore did not consider analysing daily sleep onset an appropriate measure of mouse behaviour; consistent with this, Martin-Fairey *et al.* 2019 analysed sleep onset in humans but not mice in their paper.

The changes to the manuscript are as follows:

Methods:

For clarity we have added that our metabolic cage system was not equipped with a running wheel.

“All mice were single-housed in metabolic cages (Promethion Sable System; Las Vegas, USA), equipped with a food and water hopper/scale system but no running wheel, and acclimatised for 7 days.”

[Line 141-144, page 5]

*“Unlike previous studies where activity analyses have been based on running wheel movement (Martin-Fairey *et al.*, 2019; Yaw *et al.*, 2021), we recorded spontaneous activity within the cage. We therefore developed criteria to define activity onset based on cage activity measures. We first defined a threshold activity for each individual mouse as the average of dark-phase activity during the acclimatisation period. We defined the onset of dark-phase activity for each study day as the first time at which activity exceeded this threshold for the next three consecutive 5 minute blocks. Confirming that this definition captures normal dark-phase onset behaviour in non-pregnant mice, activity onset occurred within 2 h of lights off for 95.2% of data sets collected during the acclimatisation period, including two full dark-phase cycles and all mice achieved activity onset within 2 h of lights off on at least one acclimatisation night. For mice that did not achieve activity onset within the dark-phase of any study day, activity onset time was set at 12 h for analysis.”*

[Line 192-203, page 7]

“Timing of activity onset was analysed using a mixed model, with pregnancy status (between animal factor) and study day (within animal factor), using SPSS v. 28 (IBM Corporation, Armonk, NY).”

[Line 281-283, page 10]

Results:

“Effects of pregnancy on dark-phase activity onset

The timing of activity onset is illustrated in figure 3, including examples of activity onset

*during the acclimatisation period and on day 8 of the study in non-pregnant (Fig 3 A & B respectively) and pregnant (Fig 3 C & D respectively) mice. The time after lights off at which mice achieved their individual activity threshold changed differently across the study in non-pregnant and pregnant mice (Fig 3E, day*pregnancy interaction $P = 0.042$). In non-pregnant mice, the timing of activity onset remained consistent across days (Fig 3E, $P = 0.352$), being similar during acclimatisation and during the study, as shown for an individual mouse (Fig 3A and 3B). In contrast, pregnant mice took longer to reach their activity threshold as pregnancy progressed (Fig 3E, $P < 0.001$). The example in Fig 3C and 3D shows data for an individual mouse that achieved activity onset at 40 minutes after lights off during an acclimatisation night but did not achieve its individual activity threshold at day 8 of pregnancy. Remarkably, activity onset was delayed in pregnant mice even at day 1 of pregnancy (Difference 1.86 (1.99) h, $P = 0.049$), and at day 17 of the study, activity onset was 6.07 (5.17) h later in pregnant than non-pregnant mice ($P = 0.007$)."*

[Line 365-379, page 13]

Discussion:

"In the present study, pregnant mice took longer to reach their activity threshold as pregnancy progressed, where we observed a 1.86 (1.99) h delay in the time to reach activity onset in the dark-phase on day 1, and a 6.07 (5.17) h delay on day 17, in pregnant compared to non-pregnant mice."

[Line 579-582, page 20]

Figure:

"Figure 3. The impact of pregnancy on the time of first activity onset during the dark-phase.

Daily activity onset in the dark-phase was defined as the time at which the mouse first achieved 3 consecutive, 5 minute periods of movement greater than its the average night-time activity during the acclimatisation period. Representative plots of cage activity recorded every 5 minutes are shown for acclimatisation (Panels A and C) and study day 8 (Panels B and D) for a representative non-pregnant (A and B) and pregnant (C and D) mouse during the dark-phase (shaded area). The dotted line indicates the acclimatisation average dark-phase activity for the individual mouse, and the downwards arrow shows the time at which the mouse met the criterion for activity-onset. The pregnant mouse on day 8 (D) did not reach the criterion for activity-onset. Average timing of dark-phase activity onset in non-pregnant (blue symbols and line, $N \geq 5$) and pregnant (red symbols and line, $N \geq 11$) mice is presented as mean (SD) for the acclimatisation period and for each study day (Panel E)."

[Line 778-792, page 25-26]

[Page 30]

Ib As presented, we lack a justification for treating the first, second and third weeks of pregnancy as categories during pregnancy for analysis of daily rhythms.

As discussed above, we have added analysis of daily onset of activity behaviour to the manuscript, and this has been analysed across each day of the study, not within study blocks.

Our analysis of daily behavioural peak timing and amplitude, however, required grouping data into three time periods to provide sufficient power, as discussed below. Firstly, we have replaced the phrases ‘weeks of pregnancy’ and ‘study weeks’ with ‘study blocks’, correcting a source of potential confusion – the data is divided into periods of similar duration, and these periods are not full weeks of pregnancy. It was necessary to collect tissues from our late pregnant group at day 17.5 after mating and not closer to term in order to avoid potential ethical issues if mice delivered pups and were lost to the study. Individual mouse data for

metabolic cage recordings are thus divided into three periods; block 1: day 0.5-6.5, block 2: day 6.5-12.5 and block 3: day 12.5-17.5.

Studying behaviour for three periods rather than each day was necessary to provide sufficient power for the analysis of behavioural timing and amplitude. Our model used hourly averages for each behaviour, and compared pregnant and non-pregnant mice, requiring 48 degrees of freedom for the spline analysis. If we were to analyse behaviour across each of 17 days, our spline fit would require 816 degrees of freedom. To avoid overfitting the model and too many degrees of freedom, we therefore chose to separate the data into time blocks reflective of the design of the animal experiments. The current paper used metabolic cage data previously collected for a published paper (Li *et al.*, 2021) investigating the impact of pregnancy on gastric vagal afferent function and related food intake parameters, such as meal size, duration and number. Consequently, a considerable amount of collected metabolic data was unanalysed and used for the current study. In the previous study, pregnant mice were humanely killed and tissue collected for assessment of vagal afferent function at early- (day 6.5), mid- (12.5) and late-pregnancy (17.5), our time blocks in the current study correspond to the time of tissue collection and the numbers of mice remaining at each stage.

2. Can the authors explain what is learned by "modeling" their data (e.g. Fig 1D-F; 2D-F; etc.)? The models appear to be a way of representing the same data without furthering insight into the underlying mechanisms or consequences of changes presented with the data averaged across mice.

Our modelling approach provides far richer information on behaviours during pregnancy than analysis of the timing of a single daily behavioural threshold. Our approach captures the timing and amplitude of multiple peaks in each behaviour throughout the circadian cycle and allows for assessment of when this differs between non-pregnant and pregnant mice. This approach therefore allows us to assess whether other behavioural peaks are altered during pregnancy rather than defining only the changes in onset timing of a single behavioural event, such as activity onset. We believe this approach is a novel aspect of our design and provides greater insight into the effects of pregnancy on behavioural patterns. We have added text to reflect the strengths of our modelling to the paper, as expanded below:

“Additionally, a strength of the current modelling approach is capturing the timing and amplitude of multiple peaks in each behaviour throughout the circadian cycle and assessing whether this differs between groups. This approach is a novel aspect of the statistical design and provides greater insight into the effects of pregnancy on behavioural patterns rather than single onset events.”

[line 603-607, page 20]

You will need to include the data requested in point 1 above. You should also avoid the use of "modelling" if it is providing no greater insight than that provided by the raw data, or alternatively you are able to comprehensively justify its inclusion. The Journal of Physiology's remit is to publish original research that illustrates new physiological principles or mechanisms and these requests to you will improve the granularity of the data and increase the priority for publication.

We have now included the time of activity onset data requested in point 1 above. However, we have also retained the original modelling approach along with the raw data in the revised manuscript. This modelling approach uses the raw data and captures the timing and amplitude of multiple peaks in each behaviour throughout the circadian cycle allowing for assessment of when this differs between groups. It therefore adds more information than the raw data alone. We believe this approach is a novel aspect of our design and provides greater insight into the effects of pregnancy on behavioural patterns.

Response to referee 1:

1. a) The authors claim to see changes in daily patterns that differ from prior reports (e.g., Martin-Fairey et al., 2019 and Yaw et al., 2021) but have analyzed their data in a unique way. In addition to providing hourly averages from each week of pregnancy, the paper would benefit from presenting the hourly average across mice throughout pregnancy (e.g., as in Martin-Fairey et al. 2019). b) As presented, we lack a justification for treating the first, second and third weeks of pregnancy as categories during pregnancy for analysis of daily rhythms.

Please see the response to part 1 of the editor comments.

2. Can the authors explain what is learned by "modeling" their data (e.g. Fig 1D-F; 2D-F; etc.)? The models appear to be a way of representing the same data without furthering insight into the underlying mechanisms or consequences of changes presented with the data averaged across mice. You should also avoid the use of "modelling" if it is providing no greater insight than that provided by the raw data, or alternatively you are able to comprehensively justify its inclusion.

Please see the response to part 2 of the editor comments.

References

- Hiyoshi H, Terao A, Okamatsu-Ogura Y & Kimura K. (2014). Characteristics of sleep and wakefulness in wild-derived inbred mice. *Exp Anim* **63**, 205-213.
- Li H, Clarke GS, Christie S, Ladyman SR, Kentish SJ, Young RL, Gatford KL & Page AJ. (2021). Pregnancy-related plasticity of gastric vagal afferent signals in mice. *Am J Physiol Gastrointest Liver Physiol* **320**, 183-192.
- Martin-Fairey CA, Zhao P, Wan L, Roenneberg T, Fay J, Ma X, McCarthy R, Jungheim ES, England SK & Herzog ED. (2019). Pregnancy induces an earlier chronotype in both mice and women. *J Biol Rhythms* **34**, 323-331.
- Yaw AM, Duong TV, Nguyen D & Hoffmann HM. (2021). Circadian rhythms in the mouse reproductive axis during the estrous cycle and pregnancy. *J Neurosci Res* **99**, 294-308.

Dear Dr Clarke,

Re: JP-RP-2024-285553R2-A "Circadian patterns of behaviour change during pregnancy in mice" by Georgia S Clarke, Andrew Vincent, Sharon R Ladyman, Kathryn L Gatford, and Amanda J Page

We are pleased to tell you that your paper has been accepted for publication in The Journal of Physiology.

Authors should note that it is too late at this point to offer corrections prior to proofing. Major corrections at proof stage, such as changes to figures, will be referred to the Editors for approval before they can be incorporated. Only minor changes, such as to style and consistency, should be made at proof stage. Changes that need to be made after proof stage will usually require a formal correction notice.

If you would like to receive our 'Research Roundup', a monthly newsletter highlighting the cutting-edge research published in The Physiological Society's family of journals (The Journal of Physiology, Experimental Physiology and Physiological Reports), please click this link, fill in your name and email address and select 'Research Roundup': <https://www.physoc.org/journals-and-media/membernews/>.

Yours sincerely,

Paul Greenhaff
Senior Editor
The Journal of Physiology

P.S. - You can help your research get the attention it deserves! Check out Wiley's free Promotion Guide for best-practice recommendations for promoting your work at www.wileyauthors.com/eeo/guide. You can learn more about Wiley Editing Services which offers professional video, design, and writing services to create shareable video abstracts, infographics, conference posters, lay summaries, and research news stories for your research at www.wileyauthors.com/eeo/promotion.

IMPORTANT NOTICE ABOUT OPEN ACCESS: To assist authors whose funding agencies mandate public access to published research findings sooner than 12 months after publication, The Journal of Physiology allows authors to pay an Open Access (OA) fee to have their papers made freely available immediately on publication.

You can check if your funder or institution has a Wiley Open Access Account here: <https://authorservices.wiley.com/author-resources/Journal-Authors/licensing-and-open-access/open-access/author-compliance-tool.html>.

EDITOR COMMENTS

Reviewing Editor:

Both reviewers felt that authors adequately responded to previous reviews and appreciate the additional data analysis and new Figure 3. They have no further suggestions.

Senior Editor:

Thank you for addressing the concerns previously raised and adding new data and text to the manuscript. The reviewers

and Reviewing Editor are of the opinion that the manuscript has been improved and it is now acceptable for publication. Thank you for considering The Journal of Physiology.

REFEREE COMMENTS

Referee #1:

The authors had added new analyses and text to indicate their pregnant mice showed delayed onset of daily locomotion (new Figure 3). This addresses suggestions from prior reviews. It is intriguing that food intake, water intake, and wakefulness all show an advanced onset during late pregnancy, at odds with these open field motion results. I have no further suggestions.

Referee #2:

No further comments.

2nd Confidential Review

31-Jan-2024